# Structural mechanism of laminin recognition by integrin

Takao Arimori [1], Naoyuki Miyazaki [1,2], Emiko Mihara [1], Mamoru Takizawa[3], Yukimasa Taniguchi[3], Carlos Cabañas [4,5,6], Kiyotoshi Sekiguchi[3] & Junichi Takagi [1✉]

Recognition of laminin by integrin receptors is central to the epithelial cell adhesion to basement membrane, but the structural background of this molecular interaction remained elusive. Here, we report the structures of the prototypic laminin receptor α6β1 integrin alone and in complex with three-chain laminin-511 fragment determined via crystallography and cryo-electron microscopy, respectively. The laminin-integrin interface is made up of several binding sites located on all five subunits, with the laminin γ1 chain C-terminal portion providing focal interaction using two carboxylate anchor points to bridge metal-ion dependent adhesion site of integrin β1 subunit and Asn189 of integrin α6 subunit. Laminin α5 chain also contributes to the affinity and specificity by making electrostatic interactions with large surface on the β-propeller domain of α6, part of which comprises an alternatively spliced X1 region. The propeller sheet corresponding to this region shows unusually high mobility, suggesting its unique role in ligand capture.

[1] Laboratory for Protein Synthesis and Expression, Institute for Protein Research, Osaka University, Suita, Osaka, Japan. [2] Life Science Center for Survival Dynamics, Tsukuba Advanced Research Alliance, University of Tsukuba, Ibaraki, Japan. [3] Division of Matrixome Research and Application, Institute for Protein Research, Osaka University, Suita, Osaka, Japan. [4] Cell-cell Communication & Inflammation Unit, Centro de Biología Molecular Severo Ochoa (CSIC-UAM), Madrid, Spain. [5] Department of Immunology, Ophthalmology and Otorhinolaryngology (IOO), Faculty of Medicine, Universidad Complutense de Madrid, Madrid, Spain. [6] Instituto de Investigación Sanitaria Hospital 12 Octubre (i+12), Madrid, Spain. ✉email: takagi@protein.osaka-u.ac.jp

ntegrins are one of the most important classes of receptors that mediate adhesion of cells to extracellular matrix (ECM) components, which is the hallmark of multicellular organisms. Integrins exist as a heterodimeric molecule comprising two type I transmembrane subunits α and β, yielding various α/β combinations with distinct ligand specificity[1]. For example, *Caenorhabditis elegans* has only one β subunit (*pat-3*), shared by two α subunits (*pat-2* and *ina-1*)[2], while humans have eight β and eighteen α subunits to assemble more than twenty-four heterodimeric integrins, which can be divided into four classes (Fig. 1a). Phylogenetic analysis has suggested that *pat-2* and *ina-1* have evolved into two distinct classes, the Arg-Gly-Asp (RGD)-binding class and the laminin-binding class, respectively[3]. Therefore, the laminin-binding class represents one of the most ancient integrin classes conserved throughout the metazoan evolution, playing fundamental roles in the cell attachment to the basement membrane during development[4,5].

Laminins are heterotrimeric molecules consisting of α, β, and γ chains and are major component of basement membranes. There are five α chains (α1–5), three β chains (β1–3), and three γ chains (γ1–3) in mammals, which constitute at least 16 laminin isoforms each showing some preferential expression in certain tissues/

organs[6–8]. Laminin-511 (LM511, α5, β1, and γ1 chain composition) is one of the most abundant laminin isoforms present in adult epithelial tissues, but also involved in development and is implicated in the selective differentiation of stem cells[9,10]. Intact laminin heterotrimer is a very large (>800 kDa) molecule with a cross-shaped appearance, and its integrin binding site has been narrowed down to the C-terminal one-tenth of the molecule called "E8" fragment[11]. We[12] and others[13] have succeeded in recombinantly producing E8 fragment of human LM511 and mouse LM111 (mLM111) and solved their crystal structures at 1.80 and 2.13 Å resolutions, respectively. These structures revealed the architecture of the E8 fragment, where the coiled-coil domain that bundles the α/β/γ chains and the closely-arranged three laminin globular (LG) domains (LG1-3) of the α chain form a ladle-like shape. Although we have provided evidence that the C-terminal region of the LMγ1 chain (referred to as LMγ1-tail or simply γ1-tail hereafter) and the bottom face of the LG domains conspire to form integrin binding interface, we have been unable to show that LMγ1-tail peptides can directly bind to integrins as many RGD-based peptides and mimetic compounds do, making it difficult to reach a consensus about the identity of the binding site(s).

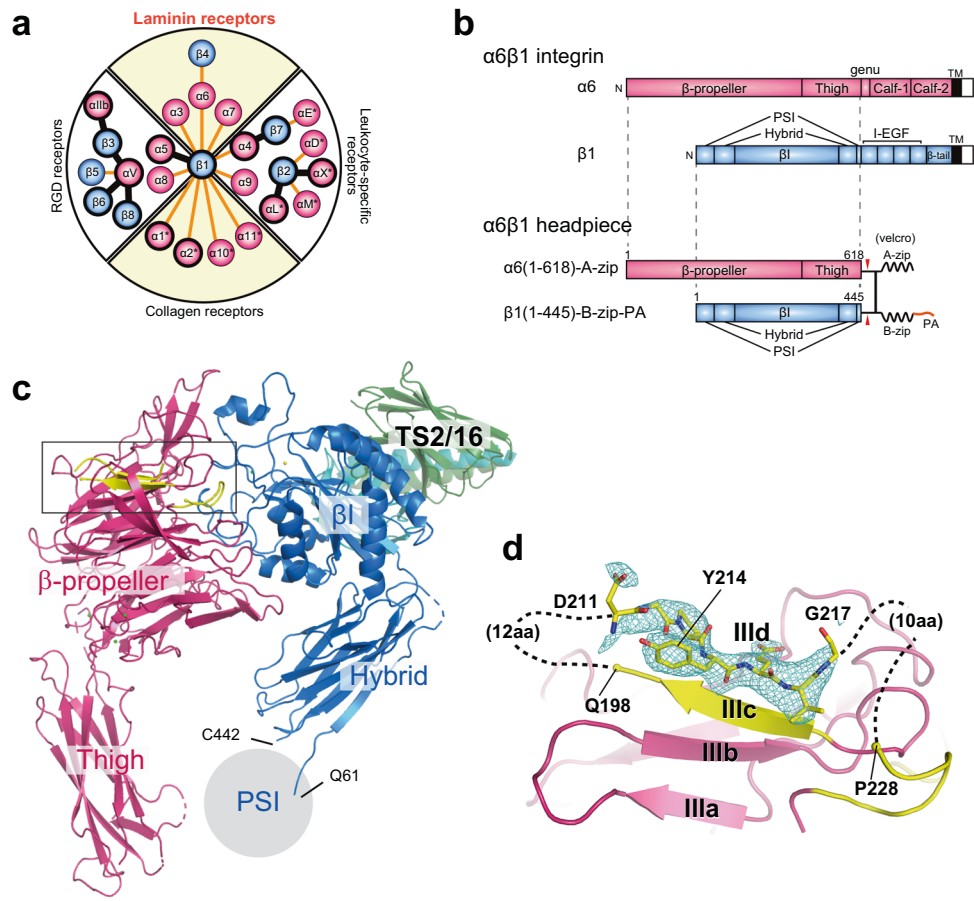

**Fig. 1 Crystal structure of the α6β1 integrin headpiece in complex with TS2/16 Fv-clasp. a** Integrin α/β pairs and classification. 18 α subunits (red circles, with the αI domain-containing ones indicated by asterisks) and 8 β subunits (blue circles) found in human forming 24 heterodimeric pairs are represented by the α-β connecting lines. Integrin subunits or heterodimers with at least partial ectodomain structures determined are denoted by thick circles. Each heterodimer falls into one of the four distinct classes. Modified from the Fig. 1 of ref. [1]. **b** Domain organization of α6β1 integrin and design of the α6β1 headpiece construct used for the structural analysis. **c** Ribbon presentation of the overall structure. Integrin α6 and β1 subunits are colored in hot pink and sky blue, respectively, and TS2/16 V_H-SARAH and TS2/16 V_L-SARAH composing TS2/16 Fv-clasp are in green and cyan, respectively. The location of the PSI domain (not included in the model) is denoted by gray circle, with the boundary residues (Q61 and C442) labeled. **d** An expanded view of the blade III β-sheet of the α6 subunit-β-propeller. The region indicated by a rectangle in (**c**) containing the X1 region (yellow) is shown with an $F_o$-$F_c$ electron density map corresponding to the outermost strand (IIId) contoured at 3.0 σ. The modeled heptapeptide [211]DGPYEVG[217] fitted in the map is also shown as stick models.

Our understanding about the mechanism of ECM ligand recognition by integrins has been advanced enormously by the structural analyses of numerous integrin heterodimers by X-ray crystallography as well as cryoelectron microscopy (cryo-EM). The prevailing concept is that the ligand provides carboxylate moiety to make direct coordination bond with the metal ion present in a site called Metal-Ion-Dependent Adhesion Site (MIDAS) conserved in all β subunits[14–17]. This in turn changes the coordination geometry of MIDAS and results in a large downward shift of the α7 helix in the βI domain, leading to the >60° swing-out of the β hybrid domain[18]. This local conformational change of integrin β facilitates global shape-shifting or "extension" of integrin heterodimer, and eventually transform the integrin into the force-transmission and signaling device that links ECM ligands and the cytoskeleton[19,20]. The global conformational change has also been observed with non-RGD integrins such as α4β7, αXβ2, and αLβ2 using low-resolution negative-stain electron microscopy (EM)[21–23]. Furthermore, the ligand-induced local conformational change at the MIDAS leading to the hybrid domain swing-out (or head-opening) has been visualized for many RGD-integrins at atomic resolution[16,18,24]. In contrast to those integrins, however, no structural information about the laminin-binding integrins is available to date (Fig. 1a, structurally determined integrins are circled with thick lines), except for the low-resolution negative-stain EM images[12,25]. Therefore, structure determination of laminin-binding integrin class before and after the binding of laminin is immensely needed to gain mechanistic understanding of this interaction fundamental to multicellular organisms.

We aimed at solving the 3D structure of laminin-integrin complex. Among the four laminin-binding integrin heterodimers, we focused on α6β1 because it is known to exhibit the highest binding affinity toward LM511[26]. The α6 subunit is widely distributed in various tissues, but when both β1 and β4 are present, α6 is preferentially paired with β4[27]. α6β1 integrin is expressed on cells of hematopoietic origin, as well as some neural cells. Interestingly, it is the major integrin isoform expressed on the surface of human stem cells, and adhesion to ECM via integrin α6 plays important roles in proliferation, migration, differentiation, and self-renewal of stem cells[28–31]. In fact, the E8 fragment from LM511 is currently used as an important substrate for the culture of embryonic stem cells and induced pluripotent stem cells[32]. As α6 integrins are considered as useful markers for cancer stem cells and also fundamentally involved in the pathogenesis and regulation of cancer[33,34], elucidation of their binding mechanism toward LM511 is of utmost medical importance.

Here, we report the crystal structure of the laminin receptor integrin α6β1 as well as the cryo-EM structure of the α6β1-LM511 complex. We succeed in visualizing the direct coordination bond between the E1607 in the LMγ1-tail and the MIDAS metal in β1 integrin, as well as many more amino acid residues involved in the interaction, suggesting that the ability of integrin to recognize basement membrane laminin is mediated by a cluster of small and potentially weak interactions that come from residues distributed across all five subunits. In addition, we find a unique feature of a loop segment in α6 β-propeller that may play important role in ligand capture.

## Results

### Crystal structure of the integrin α6β1 headpiece in complex with TS2/16 Fv-clasp.

To gain structural insights into the laminin-integrin interaction, we first determined the crystal structure of ligand binding fragment of human α6β1 (α6β1 headpiece, Fig. 1b). Crystallization was facilitated by complex formation with an anti-β1 antibody TS2/16, in a hyper-crystallizable antibody fragment

format "Fv-clasp" described previously[35]. In the 2.89 Å resolution structure, one α6β1-TS2/16 Fv-clasp complex was contained in an asymmetric unit (Supplementary Table 1). Four out of the five domains contained in the headpiece construct were resolved, while the PSI domain of the β1 subunit showed poor electron density and omitted from the final model (Fig. 1c). The β1 chain structure was essentially the same as that in the published α5β1 structures[14,36], and the resolved α6 domains (β-propeller and thigh) were highly similar to α5 and other α subunits. The α6 β-propeller domain assumes canonical 7-bladed propeller fold with the seven four-strand anti-parallel β sheets forming the domain core (Supplementary Fig. 1a). However, the two loops preceding the outermost strand in the blade III (strand IIId) were disordered. In the sequence spanning K199 to V227, only β-strand IIId, [211]DGPYEVG[217], showed traceable electron density (Fig. 1d). Density for the large Y214 residue allowed us to confidently assign the sequence to the structure register of this segment. No other published integrin β-propeller domains show such long disordered loops. Furthermore, this region coincides with the alternatively spliced "X1 region" (residues I193-L235) found only in laminin-binding α subunits[37], suggesting its possible role in ligand capture and/or specificity. This point will be discussed later in more detail.

The TS2/16 used to facilitate crystallization is an activating antibody that increases ligand binding affinity of all β1 integrins. Its binding epitope had been mapped to N207/K208/V211 located in the lower half of the α2 helix of the βI domain[38], which is confirmed by the current structure (Fig. 2a). In the crystal, the TS2/16-bound α6β1 adopts a closed conformation with the β1 hybrid domain tucked inside[39]. Furthermore, the structure of the TS2/16-bound βI domain was indistinguishable from that in α5β1 integrin structure crystallized without TS2/16 (PDB: 4wjk), showing overall RMSD of 0.37 Å for 216 Cα atoms when superposed (Fig. 2a). This clearly indicates that TS2/16 binding does not automatically induce global or local conformational changes of β1 subunit, consistent with the previous negative-stain EM results showing that it can bind to α5β1 integrin regardless of its conformational state[39].

All integrin β1 subunits contain three metal-binding sites called Synergistic Metal ion Binding Site (SyMBS), MIDAS, and ADjacent to MIDAS (ADMIDAS), each known to bind Ca²⁺, Mg²⁺, and Ca²⁺ under physiological buffer conditions. However, we could not see clear electron density for ADMIDAS metal in our structure, while electron densities were clearly visible for metals in SyMBS and MIDAS using a simulated-annealing omit map (Fig. 2b). Comparison of the current structure with the Ca²⁺-containing ADMIDAS structure of α5β1 (4wjk) indicates its incompatibility with Ca²⁺ coordination at this site due to a sidechain flip of the D137 and a ~1.8 Å shift of the A342 carbonyl (Fig. 2c), making us conclude that the ADMIDAS is unoccupied in the current α6β1 structure. Although ~0.5 mM Ca²⁺ was present in the crystal, which had been directly flash frozen before the diffraction experiments, the presence of Ca²⁺-chelating phosphate ions in the crystallization buffer may have lowered its effective concentration, causing the loss of ADMIDAS Ca²⁺.

### Structure determination of the α6β1-LM511-TS2/16-HUTS-4 quaternary complex by single-particle cryo-EM.

We next determined the structure of α6β1 headpiece in complex with its ligand LM511 by a cryo-EM approach. As for the LM511, we utilized the E8-like truncated LM511 (tLM511) design reported previously[12]. In order to maximize the stability of the complex as well as its conformational homogeneity, we utilized two antibodies in the Fv-clasp format; the affinity-enhancing TS2/16 and the conformation-stabilizing HUTS-4. The latter was included because it is known to bind inside the β1 hybrid domain in the

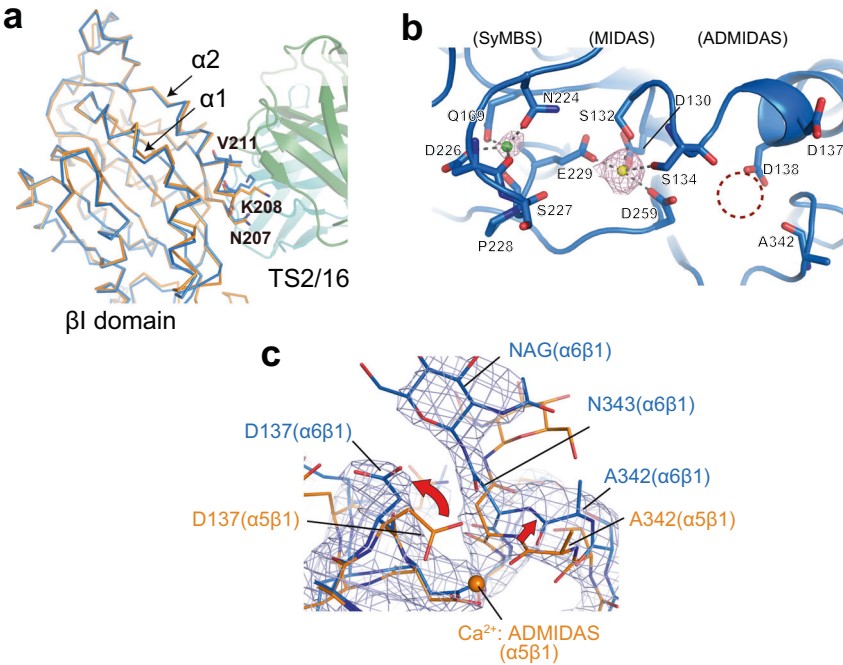

**Fig. 2 Structure of the βI domain. a** Structural comparison of β1 integrins before and after the TS2/16 binding. The β1 subunit portions from the TS2/16-bound α6β1 headpiece (sky blue) and the unbound α5β1 headpiece (PDB ID:4wjk, orange) are superposed at the βI domain and shown as Cα-tracing. TS2/16 Fv-clasp bound to α6β1 is shown as a translucent ribbon diagram. The α2 helix of βI domain harboring TS2/16 epitope residues (N207/K208/V211) and the adjacent α1 helix are labeled. **b** Metal ion coordination sites in the α6β1 headpiece structure. Simulated-annealing $F_o$-$F_c$ omit map for metal ions contoured at 2.5 σ is shown in magenta. $Ca^{2+}$ and $Mg^{2+}$ ions assigned in the SyMBS and the MIDAS are shown as green and yellow spheres, respectively. No electron density was observed at the ADMIDAS (dotted circle). **c** $2F_o$-$F_c$ electron density map contoured at 1.2 σ around the ADMIDAS region in the crystal structure of the α6β1-TS2/16 Fv-clasp complex (sky blue), superposed onto the α5β1 headpiece structure (orange) as in **a**. $Ca^{2+}$ ion bound to the α5β1 headpiece is shown as an orange sphere.

open conformation and expected to prevent closure by acting like a wedge[40]. A quaternary complex comprising α6β1 headpiece, tLM511, TS2/16 Fv-clasp, and HUTS-4 Fv-clasp (a total mass of ~290 kDa) was purified by SEC (Supplementary Notes and Supplementary Fig. 2), and cryo-EM images were collected using Titan Krios equipped with a Falcon 3EC direct electron detector. Star-shaped particles representing the quaternary complex were visible in both raw images and after 2D averaging (Supplementary Fig. 3a, b). From a total of >2 million particle images, a single-particle analysis derived a cryo-EM density map at 3.9 Å resolution (Fig. 3a and Supplementary Fig. 3). As the first step toward the reliable atomic model building, we fitted the crystal structures of each unit contained in the complex individually to the map. The structural units used for the fitting included followings; α6-β propeller (this work), α6-Thigh (this work), β1-βI (this work), β1-Hybrid (PDB: 4wk0), β1-PSI (PDB: 4wk0), LM511-coiled-coil (PDB: 5xau), LMα5-LG1 (PDB: 5xau), LMα5-LG2 (PDB: 5xau), LMα5-LG3 (PDB: 5xau), TS2/16 Fv-clasp (PDB: 5xcx), and HUTS-4 Fv-clasp (this work, Supplementary Fig. 1b). Models of the N-terminal half of the coiled-coil domain of tLM511 and a long helix located at the bottom of HUTS-4 Fv-clasp were omitted from the model due to the low quality of the corresponding cryo-EM map. All the crystal structures could be fitted very well in the map except for the βI domain of the integrin β1 subunit, where apparent discrepancies between the crystal structure and the map were observed for the α1 and α7 helices, forcing us to manually build the model. The positions of the cations and the conformation of the coordinating residues were modeled using the crystal structure of the αIIbβ3 in open conformation (PDB: 2vdo) as a guide. We assigned three $Mn^{2+}$ ions in the SyMBS, MIDAS, and ADMIDAS, following the previously reported integrin crystal structures determined under a similar

condition (i.e., in the presence of 1 mM $Mn^{2+}$)[15,18]. Despite the lack of ADMIDAS metal in the crystal structure of the ligand-unbound state (Fig. 2b), we assume that it is occupied in this complex structure based on the buffer condition (i.e., inclusion of 1 mM $Mn^{2+}$ and absence of chelating species) as well as the volume of the map at the corresponding region (Supplementary Fig. 4a). After multiple rounds of refinements, a final atomic model of the quaternary complex composed of α6β1 headpiece, tLM511, TS2/16 Fv-clasp, and HUTS-4 Fv-clasp (simply called α6β1-LM511 complex hereafter) was built (Fig. 3b, Supplemental Movie 1).

Overall, the LM511 and the α6 subunit parts showed no significant structural differences from their solitary crystal structures, indicating the lack of conformational adjustments upon the interaction (Fig. 4a, c). In contrast, the β1 subunit underwent both global and local conformational changes. First, the hybrid domain of the β1 subunit is swung out, and α6β1 adopts an open-head conformation similar to that observed in other ligand-bound β subunits (Fig. 4a)[19,41–44]. In addition, the α1 helix in the βI domain moved ~3 Å toward the MIDAS, and the α7 helix shifted downward to allow the swing-out of the hybrid domain (Fig. 4a, expanded panel). The down-shift of the α7 helix was accompanied by a large positional change of an *N*-acetyl glucosamine linked to N343 at the beginning of the α7, which was evident in the EM map (Supplementary Fig. 4b). The above configurations of the α1 and α7 helices as well as the swung-out hybrid domain seen in the α6β1-LM511 complex are very similar to those in the "open-head" conformation of ligand-bound αIIbβ3 (Supplementary Fig. 5a), indicating that the same ligand-induced conformational change mechanism found in many RGD-binding integrins also applies to laminin-binding integrins.

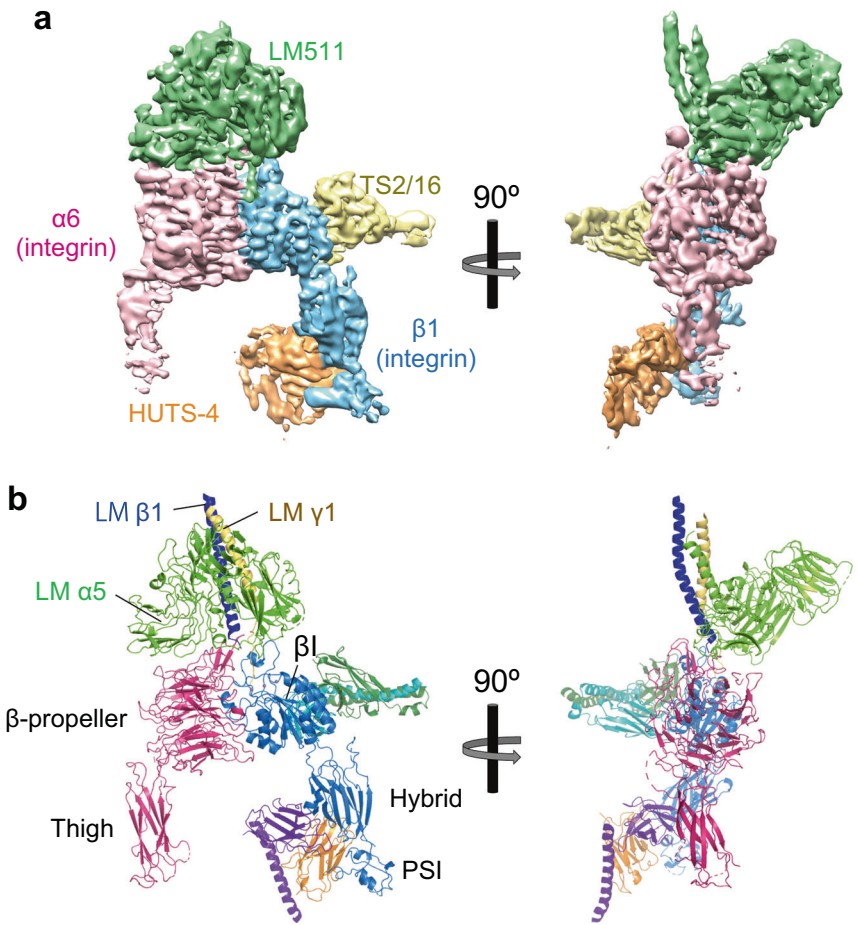

**Fig. 3 Cryo-EM structure of a quaternary complex of the α6β1 headpiece, tLM511, TS2/16 Fv-clasp, and HUTS-4 Fv-clasp. a** Cryo-EM density map. The map is colored as follows; integrin α6 subunit in pale pink, integrin β1 subunit in light blue, LM511 in pale green, TS2/16 Fv-clasp in pale yellow, and HUTS-4 Fv-clasp in pale orange. The map is drawn at the contour level of 0.05 in Chimera. **b** Ribbon diagram of the atomic model of the quaternary complex. Integrin and TS2/16 are shown in the same color scheme as Fig. 1c. Laminin α5, β1, and γ1 chains are shown in light green, blue, and yellow, respectively. $V_H$-SARAH and $V_L$-SARAH of HUTS-4 Fv-clasp are shown in orange and purple, respectively.

**Integrin-antibody interfaces**. HUTS-4 is a conformation-reporting antibody against β1 that preferentially binds to ligand-occupied β1 integrins. Consistent with previous epitope mapping result, HUTS-4 recognized species-specific residues E371 and K417 located at the inner face of the lower region of the β1 hybrid domain, explaining why it cannot bind to closed β1 integrin with tucked hybrid domain (Fig. 4b and Supplementary Fig. 5b)[45]. Unexpectedly, HUTS-4 also lightly touches the PSI domain, which may have contributed to the resolution of this domain in the cryo-EM map by fixing the PSI-hybrid inter-domain angle.

In the laminin-bound α6β1 cryo-EM structure, TS2/16 bound to βI domain using the same interface seen in the crystal structure without laminin. To our surprise, superposition of ligand unbound structure (X-ray) with the ligand bound form (EM) at the Fv portion of TS2/16 revealed that there was virtually no difference between them on either antibody or integrin side (Supplementary Fig. 5c), indicating that the TS2/16 binding does not cause conformational changes in the βI domain, at least with an extent noticeable at the resolution of the current cryo-EM structure (3.9 Å). As TS2/16 is an activating antibody, theoretical consideration of binding energetics tells us that it must bind with higher affinity to the ligand-bound (open) integrin than the unbound (closed) integrin. Therefore, we compared the binding affinity of TS2/16 toward ligand unbound and ligand-bound α5β1 integrins on K562 cell surface, in a FACS-based binding assay

using fluorescent TS2/16 monomer. In fact, the affinity of TS2/16 in the presence of $Mn^{2+}$ and RGD was ~70% higher than that in the resting condition (Supplementary Fig. 5d), confirming the theoretical prediction. More precise understanding of the mechanism of the allosteric integrin activation by TS2/16 should await further structural and functional analyses.

**Recognition of laminin β1 and γ1 chains by α6β1 integrin**. The cryo-EM structure of the quaternary complex enabled the detailed analysis of the laminin-integrin interface. tLM511 bound to the top of the α6β1 headpiece using its bottom portion where coiled-coil of three chains converge near the C-terminus of the LMβ1 chain, burying 1372.7 Å$^2$ and 1330.6 Å$^2$ of solvent accessible surfaces of LM511 and α6β1 integrin, respectively (Fig. 5a). This large interface is supported by numerous contacts on both sides (Fig. 5a). We have previously shown, via mutational studies, that a conserved acidic residue located in the C-terminal region of the laminin γ chain plays a central role in the integrin binding by directly coordinating to the MIDAS cation[12,46], although we have been unable to observe direct binding between LMγ1-tail peptides and integrins. In the cryo-EM map of the α6β1-LM511 complex, density extending from P1604 (the last LMγ1 residue that was visible in the LM511 crystal structure[12]) to penetrate into the integrin β1 subunit was observed (Fig. 5b), allowing us to successfully construct a model of the LMγ1-tail all the way to the C-

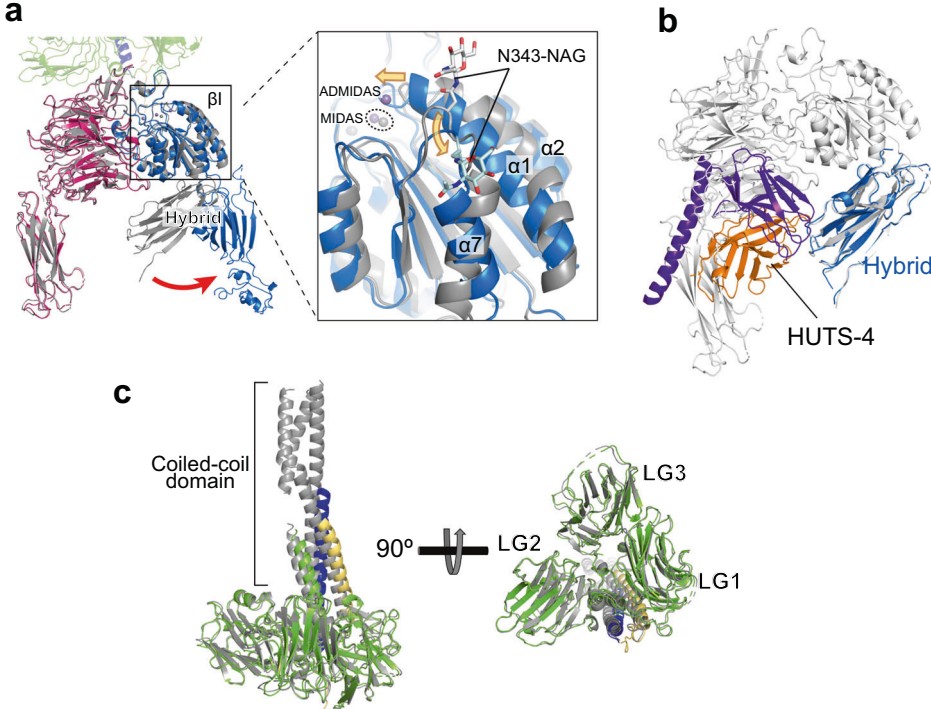

**Fig. 4 Structural changes upon ligand binding. a** Comparison of α6β1 headpiece conformations before and after ligand binding. The crystal structure of unliganded α6β1 headpiece (gray) was superposed at the βI domain onto the cryo-EM structure of the LM511 complex (color-coded as in Fig. 3b). An expanded view near the βI MIDAS (boxed) is shown in the right. The global conformational rearrangement (*i.e.*, a swing-out of the β1 hybrid domain) and the local conformational rearrangement (i.e., shifts in α1 and α7 helices) are denoted by red and orange arrows, respectively. The *N*-acetylglucosamine (NAG) residue attached to N343 is shown in stick model. **b**. HUTS-4 cannot bind to the tucked hybrid domain in the closed headpiece conformation. The hybrid domain and HUTS-4 in the cryo-EM structure (color-coded as in Fig. 3b) was superposed at the hybrid domain onto the crystal structure of closed α6β1 headpiece (gray). **c** LM511 does not undergo conformational change. The crystal structure of tLM511 (PDB ID: 5xau, gray) was superposed onto the cryo-EM structure (color-coded as in Fig. 3b) at the LG1-3 domains.

terminal P1609. Thus our structure shows that integrin binding causes γ1E1607 to become ordered, which is well positioned in the laminin sequence to form a direct coordination to the MIDAS metal (Fig. 5b). In addition to the γ1E1607-MIDAS engagement, we also found a potential hydrogen bonding interaction between the carboxyl group of the C-terminal P1609 of LMγ1 and the side chain of N189 of integrin α6, which is strictly conserved among all laminin-binding integrin α chains (Fig. 5b, e). Thus, the γ1 tail region seems to use two anchor points (E1607 and P1609) to bridge β1 and α6, in a way analogous to the subunit-bridging "R" and "D" portions of RGD peptide ligands[14–17]. Furthermore, we noted that the C-terminal residue of LMβ1, L1786, is accommodated in a small dead-end space formed by integrin α6, integrin β1, and LMγ1, likely contributing to the binding (Supplementary Fig. 6a). Since the position of this LMβ1 terminal residue relative to the γ1 chain is fixed by the γ1Cys1600-β1Cys1785 disulfide bond conserved across all the integrin-binding laminin trimers, this single-residue contact by LMβ chain may also constitute an important determinant for binding specificity.

**Recognition of laminin α5 chain by α6β1 integrin**. Compared to the focal mode of integrin binding by β1 and γ1 chains using their C-terminal ends, LMα5 uses wider surface at the bottom of LG2 domain, which is recognized exclusively by integrin α6 subunit (Fig. 5a). At the heart of this interface, a cluster of charged residues from the loop connecting blade II and III of α6 (referred to as II-III loop hereafter) point toward laminin, each of which is accompanied by residues with complementary charges on the laminin side (Fig. 5a, c), suggesting a formation of inter-molecular

electrostatic interactions. To verify the specificity of these interactions, we conducted structure-guided mutagenesis analysis. The soluble full-length ectodomain of α6β1 in which two subunits are tied by a coiled-coil motif called velcro (α6β1ec-velcro) and the N-terminally PA-tagged tLM511 (PA-tLM511) were used for immunoprecipitations (Fig. 6a). First, charge reversal mutations were introduced in the II-III loop residues to make a series of integrin α6 mutants, D153K, R155D, R157D, and K161D, and their binding toward the wild-type PA-tLM511 was evaluated. When mixtures of integrin- and laminin-expressing conditioned media were immunoprecipitated from the laminin side using anti-PA tag antibody (NZ-1)-immobilized Sepharose, only the wild-type and the K161D mutant integrins were co-precipitated (Fig. 6b, upper panel). When pulled-down from the integrin side using anti-velcro antibody (2H11)-immobilized Sepharose, binding of tLM511 again was seen only with wild-type and the K161D mutant integrins, although all mutant integrins were expressed at comparable levels (Fig. 6b, lower panel). These results suggest that D153, R155, and R157 in the II-III loop all contribute critically to the LM511 binding, while K161 contribution is marginal. We then performed a reciprocal experiment by introducing mutations in LMα5 chain to see their effect on the binding to the wild-type α6β1. As shown in Fig. 6c, mutants K3099D and D3102R showed severely decreased binding activity compared to wild-type LM511, while that of D2942R was only slightly diminished.

The same experimental setup was used to investigate the potential contribution of the N189 of integrin α6 in the recognition of the C-terminal carboxylate of LMγ1 suggested in the previous section. Thus, N189 in the context of α6β1ec-velcro was mutated to Asp residue, expecting that the negative charge

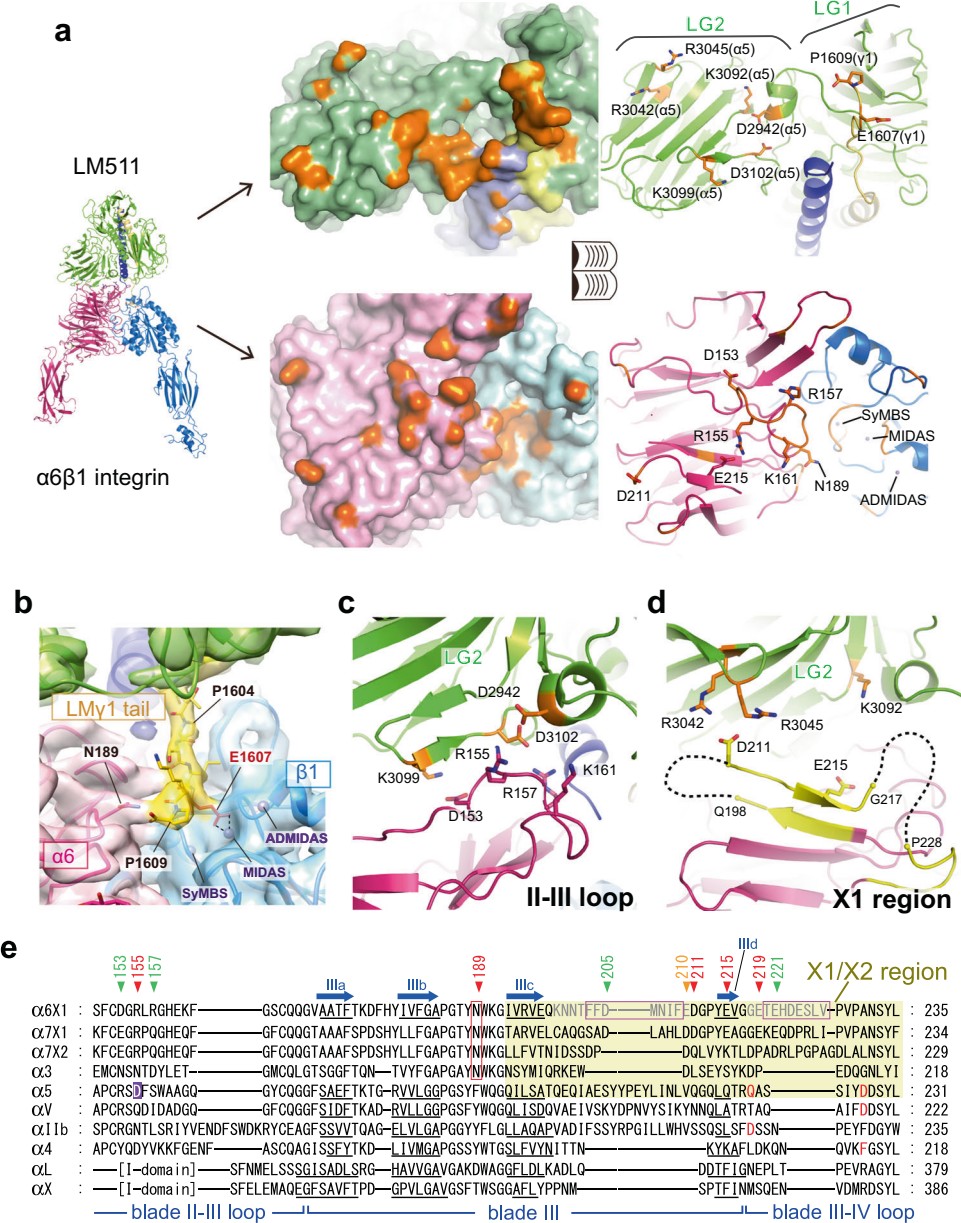

**Fig. 5 Interaction between α6β1 integrin and LM511. a** Open book view of the binding interface. (Left) Surface representations. Atoms in contact with the partner molecule (within 4 Å) are shown in orange. (Right) Cartoon representations. The key residues in this paper are shown as stick models and labeled. The surface and the cartoon figures are viewed from the same orientation. **b** Cryo-EM map near the laminin γ1-tail, drawn at the contour level of 0.05 in Chimera. The atomic model of the γ1-tail region (T1603-P1609) is shown in yellow stick model. Note that E1607 (red stick model) in the γ1-tail is in a suitable position to coordinate MIDAS cation and that the terminal carboxyl group of P1609 is pointing toward N189 of integrin α6 subunit. Close-up views for the contacts between the laminin LG2 domain and the II-III loop region **c** or the X1 region **d** of the α6 β-propeller domain. **e** Multiple sequence alignment of integrin α subunits for blade III and its flanking loop regions. Positions for the four β-strands constituting the blade III (strands IIIa-IIId) are denoted by blue arrows above the alignment, and the actual β-strand segments in each structurally determined α subunit are underlined. The Asn189 conserved in all laminin-binding integrins and the segments that are truncated in α6 mutants used in Fig. 6f are boxed in red and magenta, respectively. The α6 residues mutated for the functional study shown in Fig. 6 are labeled on top of the alignment, color-coded according to their fold reduction in the laminin affinity when mutated to Ala by red (>10-fold), orange (5–10 fold), and green (<5-fold). See also Fig. 6g.

may repel the γ1 C-terminal carboxylate without altering the geometrical shape of the α6 surface. As shown in Fig. 6b, lane 7, the N189D mutant completely lost the binding ability toward LM511. We have previously reported that shortening of the C-terminal end of LMγ1 by one or two residues led to a decreased affinity to α6β1[46]. It is thus likely that the interaction between α6 N189 and the terminal carboxyl group of LMγ1 is required for optimum binding, probably by capturing the otherwise highly

mobile LMγ1 terminus to help LMγ1-E1607 to coordinate the MIDAS cation.

As the pull-down assays described above are qualitative and have a very limited dynamic range, we next sought to establish a more quantitative assay to evaluate the contribution of each α6 residue to the overall binding, using more conservative Ala mutations. To this end, we inserted superfolder green fluorescent protein (sfGFP) between the PA tag and the truncated laminin

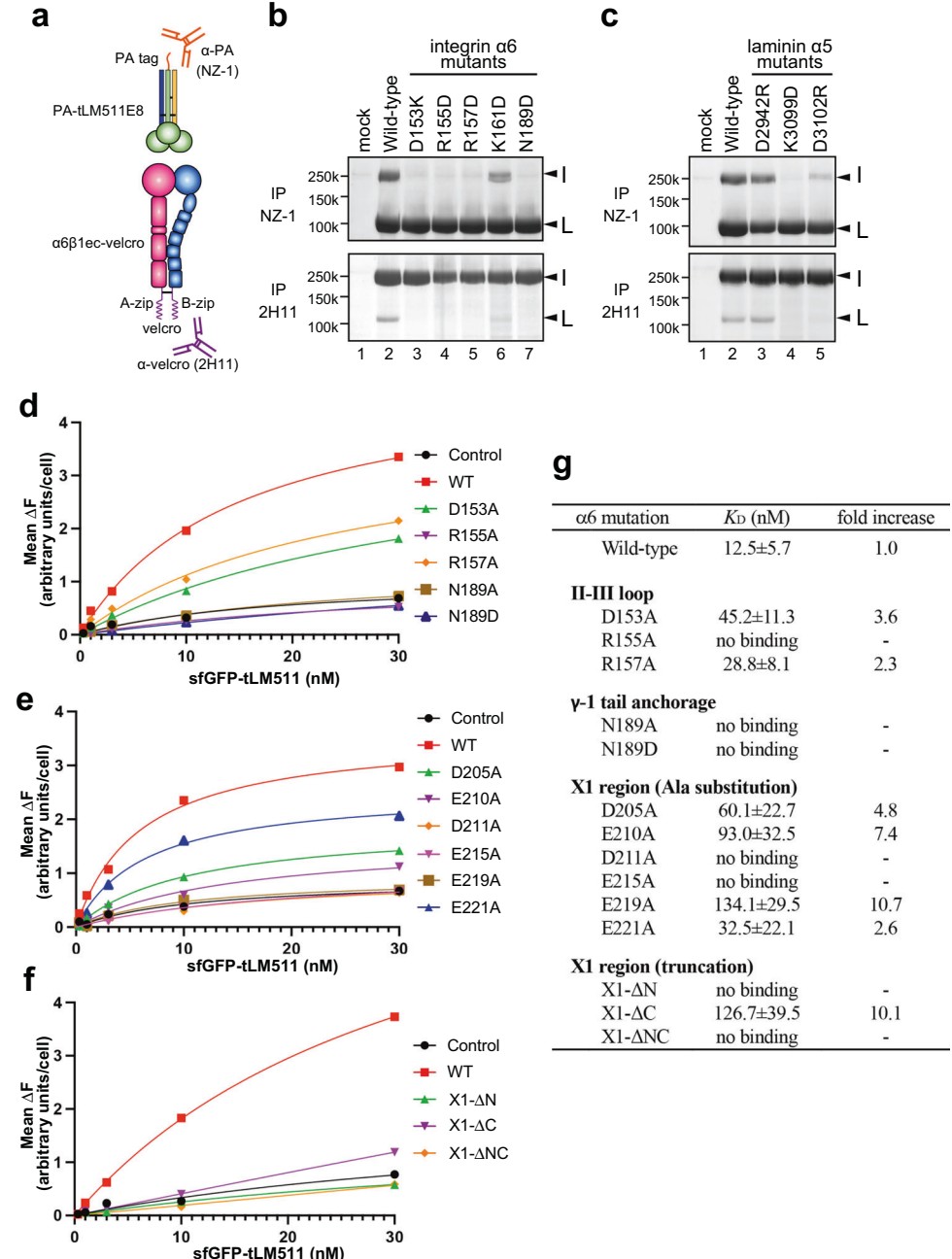

**Fig. 6 Mutagenesis of the interface residues. a** Schematics of the α6β1 integrin ectodomain (α6β1ec-velcro) and tLM511 (PA-tLM511) constructs used for the pull-down binding assay. Binding of integrin mutants to wild-type laminin **b** and laminin mutants to wild-type integrin **c**. Each construct was transfected into Expi293F cells, and culture media of each pair of integrin/laminin were mixed and subjected to immunoprecipitation using anti-PA tag antibody (NZ-1, upper panel) or anti-velcro tag antibody (2H11, lower panel), followed by SDS-PAGE analysis under non-reducing condition. Band positions of α6β1ec-velcro and PA-tLM511 are indicated by I and L, respectively. Immunoprecipitation experiments were repeated twice. Binding analysis of sfGFP-fused tLM511 to the integrin mutants expressed on the cell surface. Expi293F cells transiently expressing α6β1 integrins with the II-III loop mutations **d**, X1-region mutations **e**, and loop truncation mutations **f** were incubated with increasing concentrations of sfGFP-tLM511 to measure its binding as described in the "Method" section. Each panel shows data from a representative experiment out of three independent ones. **g** Laminin binding affinity of mutant α6β1 integrins. Equilibrium dissociation constants ($K_D$) derived from the binding isotherms like the ones shown in **d–f** by non-linear regression analysis are shown. When the binding was below the level of control non-transfected cells (black circles in **d–f**), that mutant integrin is judged as no binding. Data are mean ± SD of three independent experiments. Relative impact of each mutation on the binding is expressed as the fold-increase in the $K_D$ value from the wild-type. Uncropped gel images and source data are provided in the Source Data file.

α5 subunit in the construct described above, and the binding of resultant sfGFP-tLM511 to cells transiently expressing human α6β1 was evaluated by FACS. Concentration-dependent shift in the fluorescence histogram was observed in the presence of 0.5 mM Mn$^{2+}$ (Supplementary Fig. 7a), and the net increase of the integrated fluorescence values were plotted against the ligand concentration to derive equilibrium $K_D$ values (Fig. 6d, g). As the parent Expi293 cells express endogenous α6 subunit at low level (Supplementary Fig. 7b), binding of sfGFP-tLM511 to Expi293 (black lines in Fig. 6d–f) was regarded as a baseline. This

experiment revealed that binding to R155A mutant (purple line in Fig. 6d) was below the level of control, indicating that this residue was indeed indispensable for the laminin binding activity. In contrast, the D153A and R157A mutants retained reasonable binding ability with 3.6- and 2.3-fold increased $K_D$ values, respectively (Fig. 6g), suggesting that their contribution to the binding were partial. We also extended this assay format to evaluate the role of N189 by measuring the ligand affinity of the charge-manipulated N189D as well as more conservative N189A mutants, and found that both mutants completely lost the binding ability, confirming the importance of the N189(α6)-E1607(LMγ1) hydrogen bond.

**Potential role of the X1 region of α6 in the laminin binding.** In addition to the II-III loop residues described above, the alternatively spliced X1 region of α6 is also buried upon the binding of laminin LG2 (Fig. 5a, d). This region was highly mobile in the absence of laminin as described earlier (Fig. 1d), but upon the LM511 docking the cryo-EM map showed visible density for the two segments flanking the IIId strand (Supplementary Fig. 4c) in contrast to the total lack of electron densities in the crystal structure. Guided by the EM map, we modeled these loop segments as shown in Supplementary Fig. 4c (cyan), although they were omitted from the final model deposited in the PDB or from the Fig. 5 because the map was not clear enough at the side chain level to allow confident residue assignment. When the electrostatic potential of the top face of the integrin α6 was calculated after incorporating the putative loops, it showed highly negative surface near the X1 region (Supplementary Fig. 6b, bottom; modeled regions were bounded by dotted lines). On the laminin side, a cluster of basic residues including R3042, R3045, and K3092 reach out toward the X1 region, making highly electropositive surface (Supplementary Fig. 6b, top). Thus, the contribution of the X1 region to the interface is evident, although precise residue-wise contacts could not be visualized in the 3.9 Å cryo-EM structure. In order to investigate the role of the X1 region in more detail, we systematically mutated acidic residues in the X1 region and evaluated their ability to bind laminin using the flow cytometry quantitative assay. Although substituting the acidic residues with Ala generally lowered the affinity toward sfGFP-tLM511, the effect was partial for D205A, E210A, and E221A mutants, while D211A, E215A, and E219A mutants completely lost the activity (Fig. 6e, g). There was a trend that the impact of the Ala mutation was severer when the residue was located closer to the IIId strand (Figs. 5e and 6g). Of note, mutation of E215 in the middle of the IIId strand abolished the binding, although this residue was not even in contact with laminin in the complex structure (Fig. 5d). We then removed mobile segments flanking the IIId strand that do not bear functionally essential residues from either side (X1-ΔN or X1-ΔC) or both (X1-ΔNC), in expectation that the modifications may reduce the flexibility of the X1 region and increase the binding by stabilizing the binding-compatible conformation of the IIId strand. As the truncation was designed so as to keep the length of the loops the same as in the α3 subunit (Fig. 5e), these rather long (7–8 residues) truncations did not affect the expression level of the resultant mutant α6 (Supplementary Fig. 7b). When subjected to the tLM511 binding assay, however, none of these truncation mutants supported meaningful laminin binding (Fig. 6f), indicating that the mobile X1 region in its entirety is important for the overall binding activity.

## Discussion

For many years, it has been difficult to obtain structural information about laminin-integrin interactions at atomic detail, owing to multiple reasons including the lack of convenient ligand-mimetic compounds like RGD, difficulty in producing well-behaved and biologically active laminin fragment, and intrinsic flexibility of integrin ectodomain. We have applied several unique methodologies to overcome these hurdles and solved the structure of the integrin α6β1 headpiece in complex with the E8 fragment of LM511 by cryo-EM, which is rapidly growing as a critical method for integrin structural determination[47,48]. The complex exhibits so-called open head conformation with the β1 hybrid domain wide open, in sharp contrast to the tucked hybrid domain found in the crystal structure of the same integrin in the absence of laminin. We have observed that full-length α6β1 integrin assumes primarily extended overall conformation before the ligand binding and postulated that the linkage between the large global conformational change and the ligand affinity upregulation found in many integrins may not be applicable to laminin-binding integrins[25]. However, the current structure confirmed that the conformational rearrangement (head opening) is tightly linked to the ligand binding in α6β1 integrin as well, indicating that it is a general feature shared among major integrin classes, except for the αvβ8 integrin that is known to maintain closed conformation after the binding of protein ligand[48,49].

The structure of the complex provided the final proof for the much-debated central role of the MIDAS metal in engaging the LMγ1 E1607 carboxylate[12]. In addition to this key laminin-integrin interaction, the structure unraveled several new interface features that had not been appreciated before. First, the LMγ1-tail provides two anchor points (the MIDAS-coordinating E1607 and the α6-abutting P1609) to bridge two integrin subunits, much like the α-β bridging RGD peptide ligand. In many published structures of RGD-binding integrins in complex with RGD-like ligands, the "D" in the ligand invariably serves as the primary anchor point to engage MIDAS-bound cation, while the "R" or equivalent basic residue at the -2 position is often bound by specific residue(s) in the α subunit (e.g., α5-Q221/D227, αV-D218, and αIIb-D224, denoted by red in Fig. 5e), although there are few exceptions to this rule[50]. In α6β1-LM511 complex, the primary anchor point is the MIDAS-coordinating E1607 in the LMγ1-tail, and the second anchor is formed between the terminal carboxylate of P1609 at +2 position and the conserved N189 of α6. Interestingly, we note that RGD-binding α subunits all have Tyr/Phe at the position equivalent to α6 N189 (Fig. 5e), and they invariably contribute to the ligand binding by contacting with the aliphatic portion of the Arg. Also, the corresponding Tyr187 (Fig. 5e) in α4 plays important role in the recognition of α4-specific compound RO0505376 by making ring stacking with the third aromatic moiety of the compound[21]. In contrast, the corresponding residue in I-domain integrins are usually small amino acids (Fig. 5e), which may be explained by the lack of the second anchor point in their ligand, the internal peptide connecting the I-domain to the β-propeller[51]. Thus, the subunit-bridging, two-point anchoring mode seems to be conserved among ligands of non-I domain integrins, although the direction of the tripeptide portion is reversed in LM compared to the RGD ligands.

Another important interface concerns the blade II-III loop in the integrin α subunits' β-propeller domain. In the complex, the bottom face of the LMα5 LG2 domain docks onto this loop with a high electrostatic complementarity. The conformation of this loop in the apo form is nearly identical to that of the complex, suggesting its significant contribution to the binding energy by presenting a preformed LM docking platform at the top of the α6 subunit. On the laminin side, a set of charged residues at the edge of the LG2 domain are responsible for accepting these residues (Fig. 5c). Interestingly, these residues are well conserved in LMα1 chain in LM111 (Supplementary Fig. 8), which is a good adhesion substrate for α6β1[26]. Therefore, we predict that the

II-III loop plays similar role in recognizing LM111. We also note that the importance of the II-III loop in the recognition of protein ligands may be generalized to most if not all integrin α subunits. In I-domain integrins, for example, it is the loop harboring the ligand-binding I-domain, which can be considered as an internal protein ligand[52]. In αIIb integrin, the II-III loop forms so-called cap subdomain unique to this subunit, which has been not only structurally shown to make direct contact with a fibrinogen γ peptide[16,24], but also predicted to mediate more extensive interaction with the fibrinogen protein[53]. Furthermore, this loop contains D154 in integrin α5 (highlighted purple in Fig. 5e), which has been shown to play a key role during the interaction with the fibronectin ligand by accepting R1379 on the synergy site located in the 9th FnIII repeats[14]. Although there is no experimental structure available for the α4 or α9 integrins bound to protein ligands (e.g., VCAM-1), this region may be a good place to start a mutational analysis experiment to map the key residues responsible for the high affinity binding of protein ligands.

The alternatively spliced X1/X2 region has long been postulated to be important for ligand binding in laminin binding integrins, since in α7 the alternatively spliced versions exhibit distinct specificity against different laminins[54]. The X1 sequence of α6 is highly conserved with that of the X1 variant of α7 subunit including the acidic residues important for the binding, but differs substantially with the alternatively spliced α7 X2 variant including the overall length (Fig. 5e). It is known that α6 does not have X2 variant, although a longer variant containing extra insertion of X2 after the X1 (α6X1X2) is expressed as a minor species[55]. Interestingly, corresponding region of α3 subunit is much shorter. Our structure directly confirmed the critical involvement of this region in the ligand binding of α6X1β1 integrin by showing that the X1 loop is in fact located in the interface. However, the way α6 X1 region contributes to the laminin binding may not follow the classical lock-and-key type of protein recognition mechanism, because the sidechain conformation of the critical interface residues identified by Ala mutagenesis experiments (e.g., E210 and E219) remained ambiguous even after the ligand docking. Considering the fact that these residues can be crosslinked to basic residues in the LM5 LG2 domain when they were mutated to Cys[56], they may make direct but transient contact with laminin. Based on these considerations, it is tempting to speculate that the long and mobile X1 region is specialized in facilitating the long-range electrostatic capture of ligands, like in a fly fishing. We propose a hypothetical model of the laminin recognition mechanism by α6β1 in Fig. 7. First, the long-range ligand search via the mobile and charged X1

region will bring the bottom face of the LG1-3 into close proximity to integrin, followed by the II-III loop engagement to fix the relative orientation between the integrin and the LG trimer. Then the LMγ1-tail penetrates into the groove between the α6 and β1 subunits and needs to sample only small space to find its target, the MIDAS metal. The twin engagement via LMγ1 P1609-α6 N189 and LMγ1 E1607-MIDAS completes the local conformational rearrangement of α1 and α7 helices in the βI domain, which makes the hybrid domain to swing out. Eventually, α6β1 is converted into the extended-open conformation, and establishes the firm linkage between the basement membrane and the cytoskeleton of epithelial cells.

Structures of αVβ6 or αVβ8 integrin headpiece in complex with a protein ligand L-TGF-β have been solved by X-ray crystallography and cryo-EM, respectively, where the ligand recognitions were mediated primarily via β subunit[43,48]. The current study presents another structural framework for the interaction between integrin and its large ECM ligand, and shows how and to what extent the integrin α subunit contributes to the laminin recognition. It also makes it clear that structural analysis of integrin bound by a large protein ligand is critical in delineating the full picture of ECM recognition by integrins. Integrin research has been enormously propelled by the discovery of the RGD tripeptide in 1984[57], which marked an epoch in the history of cell adhesion research. However, the unusual success of the RGD peptide may have caused a misconception in the field that protein-protein interfaces can often be narrowed down to a linear peptide sequence. The ECM component laminin uses multiple stretches of amino acid sequences distributed widely in the structure to formulate complicated 3D pharmacophore, in order to secure the specificity and high affinity toward specific integrin receptors. This feature, at least in the case of laminin, has prevented us from devising a small molecule capable of mimicking or inhibiting the interaction. However, we hope that the structural information provided here, particularly that of the core binding interface near the LMγ1-tail, may help designing useful chemical tools to further our understanding of the cell-basement membrane interactions.

## Methods

**Residue numbering scheme.** Residue numbering for integrin subunits are based on the mature protein sequence following the convention in integrin research literatures, while laminin subunits are numbered starting with the initiation Met as residue 1.

**Sample preparation for the structural analysis.** The integrin α6β1 headpiece and tLM511 used for the structural analysis were prepared as previously described[12,35]. For production of the α6β1 headpiece fragment, coiled-coil motifs called A-zip and

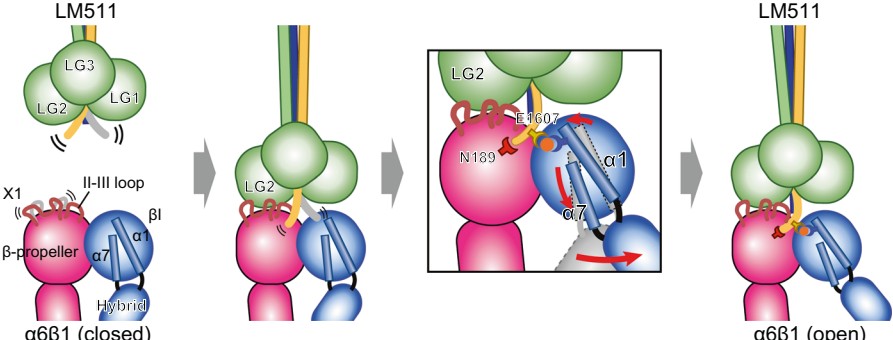

**Fig. 7 Hypothetical model of the LM511 recognition by α6β1 integrin.** In the first step, a long-range electrostatic search via the mobile and charged X1 region attracts the LG domain of laminin to integrin (the first panel). The II-III loop fixes the relative orientation between the integrin and laminin (the second panel). The flexible γ1-tail is fixed via two anchor points (i.e., LMγ1 P1609-α6 N189 and LMγ1 E1607-MIDAS), which induces local conformational rearrangement of α1 and α7 helices in the integrin βI domain, making the hybrid domain swing out (the third panel). Eventually, α6β1 is converted into the extended-open conformation (the last panel).

B-zip were appended to the C-termini of a truncated α6 (residues 1-618) and a truncated β1 (residues 1-445), respectively, to stabilize the heterodimer. A 12-residue PA tag was also appended to the C-terminus of the β1 construct to facilitate purification. After the anti-PA-tag antibody (NZ-1) immunoaffinity purification, the tag regions were removed by TEV protease digestion (the cleavage positions are indicated by red arrowheads in Fig. 1b) prior to the final purification on a Superdex 200 Increase 10/300 GL column (Cytiva) equilibrated with 20 mM Tris, 150 mM NaCl, pH 7.5 (TBS). For the tLM511 production, hexahistidine-tagged tLMα5 (residues E2655–A3327), HA-tagged tLMβ1 (residues D1714–L1786), and FLAG-tagged tLMγ1 (residues D1528–P1609) were co-transfected into FreeStyle™ 293 cells (Thermo Fisher Scientific) according to manufacturer's instruction, and the resultant tLM511 protein was purified from the conditioned media through two successive affinity chromatographies using cOmplete His-Tag Purification Resin (Roche) and DDDDK-tagged Protein PURIFICATION GEL (MBL, Nagoya, Japan), followed by a final purification on a Superdex 200 Increase 10/300 GL column (Cytiva). Anti-β1 integrin antibodies were produced in the Fv-clasp format as follows. Briefly, $V_H$ (residues 1-113) and $V_L$ (residues 1-108) genes of HUTS-4 were obtained by RT-PCR cloning from the hybridoma cells[40], while the $V_H$ and $V_L$ sequences for TS2/16 have been reported[35]. The bacterial expression plasmids for $V_H$-SARAH and $V_L$-SARAH were constructed by using pET11c. Cys87 in the $V_L$ domain of HUTS-4, a free Cys residue, was substituted with Tyr by a Quick-Change mutagenesis. Method for expression, refolding, and purification of the TS2/16 and HUTS-4 Fv-clasps were essentially the same as that described previously[35].

**Size exclusion chromatography (SEC) analysis.** tLM511, TS2/16 Fv-clasp, and/or HUTS-4 Fv-clasp were mixed with the integrin α6β1 headpiece at 1.5-fold molar excess. After 1-hour incubation at room temperature, the samples were subjected to SEC on a Superdex 200 Increase 10/300 GL column (Cytiva, 28990944) equilibrated with TBS containing 1 mM $MgCl_2$ and 1 mM $CaCl_2$ or TBS containing 1 mM $MnCl_2$ and 0.1 mM $CaCl_2$. The peak top fraction sample obtained from the SEC analysis of the α6β1 headpiece, tLM511, TS2/16 Fv-clasp, and HUTS-4 Fv-clasp mixture eluted in the 1 mM $MnCl_2$/0.1 mM $CaCl_2$ condition (indicated by a red asterisk in Supplementary Fig. 2b) was analyzed by 5-20% gradient SDS-PAGE, followed by staining with Coomassie Brilliant Blue (CBB).

**Crystallization.** For the crystallization of the α6β1 headpiece-TS2/16 Fv-clasp complex, α6β1 headpiece was mixed with TS2/16 Fv-clasp at 1.5-fold molar excess and subjected to SEC on a Superdex 200 Increase 10/300 GL column equilibrated with TBS containing 1 mM $MgCl_2$ and 1 mM $CaCl_2$. The purified sample was concentrated to 8.3 mg/ml by ultrafiltration using Amicon ultra (Millipore, 50 kDa cutoff, UFC505024). For the crystallization of HUTS-4 Fv-clasp, the purified HUTS-4 Fv-clasp sample was also concentrated to 10 mg/ml using Spin-X UF (Corning, 30 kDa cutoff, 431484). Crystallization screening was carried out by the sitting-drop vapor diffusion method at 20 °C using The Classics Suite (Qiagen, 130701), Wizard Classic 1&2 (Rigaku, 1009530 and 1009531), and ProPlex (Molecular Dimensions, MD1-38) crystallization kits. Crystals of the α6β1 headpiece-TS2/16 Fv-clasp complex appeared in a few drops, and the crystallization conditions were optimized using hanging drop vapor diffusion method at 20 °C. The diffraction quality crystals were obtained under the condition of 23% PEG1000, 0.2 M NaCl, 0.1 M Na/K phosphate, pH 6.5. Crystals of the HUTS-4 Fv-clasp were appeared under ~20 conditions in the initial screening. A crystal obtained under the condition of 1.26 M ammonium sulfate, 0.1 M MES, pH 6.0 (Wizard 2-No. 45) in the screening plate was used for data collection.

**X-ray diffraction and structure determination.** Crystals of the α6β1 headpiece-TS2/16 Fv-clasp complex were harvested and flash-frozen directly in liquid nitrogen, while crystals of the HUTS-4 Fv-clasp were transferred to well solution supplemented with 20% (v/v) glycerol before freezing. Diffraction data for the α6β1 headpiece-TS2/16 Fv-clasp complex and HUTS-4 Fv-clasp were collected at 100 K at beamlines TPS BL05A of National Synchrotron Radiation Research Center (Hsinchu, Taiwan) and BL-17A of Photon Factory (Tsukuba, Japan), respectively. Diffraction data were processed and scaled using XDS[58]. Initial phases were determined by molecular replacement method with PHASER[59] from the CCP4 package[60]. PDB IDs of search models used for the molecular replacement were 4wk0, 3vi3, and 5xcx for the α6β1 headpiece-TS2/16 Fv-clasp complex and 3qq9, 4kaq, and 5xct for HUTS-4 Fv-clasp. Structure refinements were carried out using Phenix[61], and manual model modifications were performed periodically using COOT[62]. Data collection statistics and refinement parameters are given in Supplementary Table 1.

**Cryo-EM data collection and processing.** For the cryo-EM experiments, α6β1 headpiece, tLM511, TS2/16 Fv-clasp, and HUTS-4 Fv-clasp were mixed in a molar ratio of 1:1.5:1.5:1.5 and purified on a Superdex 200 Increase 10/300 GL column equilibrated with TBS containing 1 mM $MnCl_2$ and 0.1 mM $CaCl_2$. A 2.5 μl of the sample solution of the quaternary complex (70 μg/ml) was applied to glow-discharged Quantifoil holey carbon grids (Quantifoil R2/1, Mo 300 mesh) covered with a thin, amorphous carbon film of thickness 5–10 nm. The grids were incubated in the Vitrobot Mark IV (Thermo Fisher Scientific) at 4 °C and 100% humidity for 30 s. After blotted with filter papers for 3 s, the grids were immediately plunged into liquid ethane and then transferred to a cryo-electron microscope (Titan Krios, Thermo Fisher Scientific) incorporating a field emission gun, a Cs corrector (CEOS, GmbH), a Volta phase plate, and a direct electron detection camera (Falcon 3EC, Thermo Fisher Scientific). The microscope was operated at 300 kV with a nominal magnification of ×59,000, which resulted in a calibrated pixel size of 1.113 Å. Volta phase contrast movies were recorded on the Falcon 3EC camera in linear mode as described previously[63]. Each exposure of 2 s was fractionated into 26 movie frames, leading to a total electron dose of 20 e⁻/Å², with a nominal defocus ranging from -0.6 to -0.8 μm (Supplementary Table 2). Two data sets, data set #1 (4,155 movies) and #2 (3,613 movies), were collected with EPU (Thermo Fisher Scientific) in the same experimental condition.

The workflow of cryo-EM reconstruction was summarized in Supplementary Fig. 3. Movie frames were aligned and summed using MotionCor2 software[64] to obtain a dose-weighted and motion-corrected image. Estimation of the contrast transfer function (CTF) was performed using the Gctf software[65]. The following image analysis was performed using the RELION 3.0 software package[66]. A total of 1,540,870 particles (data set #1) and 1,119,413 particles (data set #2) were automatically picked from 4155 and 3613 movies, respectively, and then they were separately used for reference-free 2D classification. Particles in the good 2D classes (827,006 particles in data set #1 and 693,594 particles in data set #2) were subjected to first round of 3D classification. The best class with 2 Fv-clasps bound in two data sets was selected (class ii in data set #1 and class I in data set #2), and then a total of 537,875 particles were combined and subjected to second round of 3D classification focused on a region at the molecular interface between the laminin and the integrin (Supplementary Fig. 3). Because of the flexibility of the thigh domain in the α subunit and the hybrid domain in the β subunit and the incomplete binding of HUTS-4 Fv-clasp, these regions were omitted from the 3D classification and refinement mask. The best class (class 3: 429,521 particles) was selected and used for 3D refinement and post-processing, which yielded a map at a 3.9 Å resolution (map A). To improve the density map at the laminin-integrin interface, we attempted further 3D focused classification and refinement. After density subtraction of the flexible parts of the thigh domain in the α6 subunit and the hybrid domain in the β1 subunit of the integrin together with the HUTS-4 Fv-clasp, images were subjected to further two rounds (3rd and 4th) of 3D classification on the laminin-integrin interface. Finally, the best class (class C: 157,515 particles) was selected and used for 3D refinement and post-processing. However, this process did not improve the resolution but rather worsened it (Supplementary Fig. 3c, map B, 4.3-Å resolution). Thus, we used the 3.9-Å resolution map (map A) for the atomic model building and structure interpretation.

**Modeling of the α6β1 headpiece-tLM511 complex.** Initial atomic model of the α6β1 headpiece-tLM511 complex was built by fitting the crystal structures described in the Results section as rigid bodies to the cryo-EM map using UCSF Chimera[67]. The atomic model of the LMγ1-tail region was built de novo in COOT[62]. The models were manually adjusted using COOT to fit to the cryo-EM map, and refined using the real-space refinement in Phenix[61]. As for the positions of the metals and the conformation of the coordinating residues in the integrin βI domain, the model was built by referring to the structure of αIIbβ3 in the open-head conformation (PDB: 2vdo) because of the indecisive map. The final model was validated using MolProbity[68], and refinement and validation statistics are summarized in Supplementary Table 2. All structural figures were prepared by PyMOL software (Schrödinger, LLC) unless otherwise specified.

**Immunoprecipitation.** Expression constructs for the soluble integrin α6β1 ecto-domain were prepared as previously described[69]. Briefly, the A-zip and the B-zip peptides that form a disulfide-linked coiled-coil called velcro were appended at the C-termini of α6 ectodomain (authentic signal sequence followed by residues 1-988) and β1 ectodomain (authentic signal sequence followed by residues 1-708), respectively, and cloned into the same pcDNA3.1-based vector. For the PA-tagged tLM511 construction, synthetic DNAs coding for tLMα5 (residues E2655–A3327), tLMβ1 (residues D1714–L1786), and tLMγ1 (residues D1528–P1609) were used. A PA tag was appended at the N-terminus of the tLMα5. These three DNA segments were individually cloned into a pcDNA3.1-based vector containing a prolactin signal sequence. All single-residue mutations were introduced by QuickChange strategy. Each of the integrin and laminin samples were transiently expressed using the Expi293 expression system (Thermo Fisher Scientific) according to the method provided by the manufacturer. Culture supernatants were harvested 4 days after the transfection, and appropriate combinations of integrin and laminin culture media were mixed and subjected to co-immunoprecipitation using anti-PA tag antibody (NZ-1)-immobilized Sepharose or anti-velcro tag antibody (2H11)-immobilized Sepharose. After washing with TBS containing 1 mM $MgCl_2$ and 1 mM $CaCl_2$, the bound samples were eluted with SDS-PAGE sample buffer and subjected to 10% SDS-PAGE analysis under non-reducing condition, followed by CBB staining.

**Preparations of fluorescently labeled TS2/16 Fv-clasp and tLM511.** To be used in the quantitative binding analysis to cell surface integrins, TS2/16 Fv-clasp and tLM511 were fluorescently labeled by fusing with superfolder green fluorescent protein (sfGFP). For making TS2/16 Fv-clasp-sfGFP, both $V_H$-SARAH and $V_L$-

SARAH coding regions were cloned in-frame into a mammalian expression vector containing mouse nidogen-1 signal sequence, and sfGFP gene (GenBank accession ASL68970.1) and hexahistidine tag were appended at the C-terminal of $V_H$-SARAH and $V_L$-SARAH, respectively. These plasmids were co-transfected into Expi293F cells and the secreted fusion protein was purified from the culture supernatants using Ni-NTA-agaraose chromatography. For making fluorescent tLM511 fragment, sfGFP gene was inserted after the PA tag in the PA-tLMα5 construct described in the previous section, and the resultant PA-sfGFP-tLMα5 plasmid was co-expressed with β1 and γ1 constructs using the Expi293F cells. The trimeric fusion protein PA-sfGFP-tLM511 was purified from the culture supernatants using NZ-1-immobilized Sepharose using the protocol described previously[70]. For both sfGFP fusions, protein concentrations were determined fluorometrically with the NanoDrop 3300 fluorospectrometer (Thermo Fisher Scientific), using the purified sfGFP as a standard.

**Flow cytometry-based binding assays.** For the quantitative binding analysis of TS2/16 to cell surface β1 integrin, K562 cells were suspended in 20 mM Hepes, 150 mM NaCl, pH 7.5, containing 0.1% BSA (HBS-BSA) at $2 \times 10^5$ cells/ml and incubated with varying concentrations of TS2/16 Fv-clasp-sfGFP in the presence of 1 mM $CaCl_2$ and 1 mM $MgCl_2$ (for ligand unbound state) or 0.5 mM $MnCl_2$, and 1 mM GRGDSP peptide (for ligand-bound state). After the incubation at room temperature for 2 h, the cells were directly subjected to flow cytometry on an EC800 system (Sony) without any washing steps. The acquired histogram data were used to extract fluorescence signal associated with each cell in a linear scale, and values from all gated cells (>5000 cells/histogram) were used to obtain mean cellular fluorescence in an arbitrary unit. Background fluorescence obtained in the absence of fluorescent protein (generally ~0.5 arbitrary units /cell) were subtracted from these values and plotted against the concentration of TS2/16, and the data were analyzed using PRISM software (ver 9.1.0, GraphPad Software, LLC.) to calculate $K_D$ values. For the quantitative analysis of tLM511 binding to various α6 mutant integrins, Expi293F cells in suspension culture were transiently transfected with various mutant versions of full-length human α6 together with wild-type β1 integrin subunits using the protocol recommended by the manufacturer. After 2 days post transfection, the cells were suspended in HBS-BSA containing 0.5 mM $MnCl_2$ and incubated with varying concentrations of sfGFP-tLM511 at room temperature for 2 h, and directly subjected to flow cytometry without washing steps. The analysis of the binding data and calculation of the $K_D$ values were performed as in the case of TS2/16 binding, although the background fluorescence of the Expi293F cells were higher than K562 cells (actual values shown in Supplementary Fig. 7a).

**Reporting summary**. Further information on research design is available in the Nature Research Reporting Summary linked to this article.

## Data availability

Atomic coordinates of HUTS4 Fv-clasp, integrin α6β1-TS2/16 Fv-clasp complex, and integrin α6β1-LM511-TS2/16 Fv-clasp-HUTS4 Fv-clasp complex have been deposited in the Protein Data Bank (PDB) with accession codes 7CEA, 7CEB, and 7CEC, respectively. Cryo-EM data of integrin α6β1-LM511-TS2/16 Fv-clasp-HUTS4 Fv-clasp complex have been deposited in the EM Data Bank (EMDB) with accession code EMD-30342 . PDB IDs of search models used for the molecular replacement are 4wk0, 3vi3, and 5xcx for the α6β1 headpiece-TS2/16 Fv-clasp complex and 3qq9, 4kaq, and 5xct for HUTS-4 Fv-clasp. GenBank accession code for sfGFP gene is ASL68970.1. All the data supporting the findings of this study are available within the article and its Supplementary Information files, and from the corresponding author upon reasonable request. Source data are provided with this paper.

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

## Acknowledgements

We would like to thank Yukinari Kato and Mika Kato-Kaneko for performing molecular cloning of HUTS-4 antibody, Kenji Iwasaki for the support of the cryo-EM data acquisition and analyses, and the staff of the beamlines at Photon Factory and National Synchrotron Radiation Research Center for their help with X-ray data collection. This work was supported in part by JSPS KAKENHI grant number JP18H02389 from Japan Society for the Promotion of Science to T.A., and by the Platform Project for Supporting Drug Discovery and Life Science Research (Basis for Innovative Drug Discovery and Life Science Research (BINDS)) funded by Japan Agency for Medical Research and Development (AMED) under Grant Number JP19am0101075 to J.T. The data collection and analyses were performed using the Molecular Cryo-Electron Microscope facility of Institute for Protein Research, Osaka University.

## Author contributions

T.A. designed and performed experiments, analyzed the data and wrote the manuscript. N.M. performed experiments, analyzed the data, and wrote the manuscript. E.M., M.T., Y.T., and C.C. performed experiments, and analyzed the data. K.S. analyzed the data and wrote the manuscript. J.T. conceived the experimental design, analyzed the data and wrote the manuscript. All authors contributed to the preparation of the manuscript.

## Competing interests

The authors declare no competing interests.
