## [Peer Review File · Nature Communications]

REVIEWER COMMENTS

Reviewer #1 (Remarks to the Author):

This is an outstanding piece of work that finally describes the structural basis for the binding of the binding of integrin alpha6 beta 1 to laminin 511. The authors employed both crystallography and cryo EM to solve the structures. The work is technically outstanding and there are no major criticisms of the paper. It has solved a longstanding problem in the field of integrin biology and is therefore highly significant to all of us who work in this field.

There are a few minor comments that would help to improve the paper.

- 1) While the English is very good there are some grammatical errors that should be fixed prior to publication.
- 2) In line 138 the authors state "Interestingly, this shape-shifting is accompanied by a side-chain flipping of the D137, which, together with a ~ 1.7 Å move of the A342 carbonyl, results in the disruption of the coordination environment of the ADjacent to MIDAS (ADMIDAS) metal (Fig. 1e). It would be nice to show a supplemental figure of the density map and how the relevant sidechains fit as their resolution is only 2.5 to 2.9 Å.
- 2) Space permitting by the editors it would be nice to show supplemental figure 4 in the main body as it is important and interesting data.
- 3) Panel A in supplemental figure shows very dense particles. It would be nice to see one where the particles are more dilute.
- 4) There appears to be an error in line 77 where figure 1A is referenced. This should be fixed

Reviewer #2 (Remarks to the Author):

This paper describes the structure of the laminin-integrin $\alpha 6\beta 1$ complex by cryo-EM and crystallography. Although numerous integrin complexes have previously been determined, there is no structural information on how laminins bind integrins. The work described includes construct optimization, conformation-stabilizing antibodies, X-ray structures of components including the conformation-stabilizing Fv-clasp construct of antibody HUTS4 and a complex of the integrin $\alpha 6\beta 1$ headpiece bound to TS2/16 Fv-clasp. These components were then assembled with a fragment of laminin (LM511) to form a high-affinity complex that was suitable for cryo-EM structure determination. This heroic effort was required to make a landmark achievement, the first molecular view of how integrins recognize laminin. However, the impact of this achievement is offset by many distractions. These include the way the paper is written, how the subject is introduced, too much emphasis on all the dead ends and hard work, much description of what did not work, tendentious claims including controversy on integrin head opening and how integrins work, over-emphasis on RGD-binding integrins, which ignores important achievements to date on how non-RGD binding integrins work, and a few doubtful interpretations that are really not necessary to the main thrust of the paper. Furthermore, the figures poorly illustrate the main point of how laminin is bound and do not clearly illustrate the overall binding site.

The overall work overcomes numerous obstacles to previous efforts on this complex, although no specific approach by itself is novel. The cryo-EM structure is intriguing and reveals a major footprint of LM511 on the $\alpha 6$ integrin subunit, but its size is not quantitated, the contacts could be put into better context, and their relative importance is not measured quantitatively. One wonders why the EM resolution was only 3.9Å, whereas the group of Yifan Chen was able to achieve substantially higher resolution on the complex of the entire $\alpha V\beta 8$ ectodomain bound to proTGF- $\beta 1$, an even larger complex.

The $\alpha V\beta 8$ structure, however, focused refinement only on the center of the complex. Whether this was done here is not defined. The paper would be greatly improved if the EM was at higher resolution and

details were better described.

The authors need to put much more work into carefully interpreting the structure, their conclusions and writing about what is novel in their structure. And spend less time on points that are peripheral to their point on laminin. The previous laminin crystal structures already made clear that all three chains come closely together at the key Asp sidechain that is recognized, and must all be required for integrin recognition, so there is really no need to stress the distinction from RGD. Yes, lots of other people tried to follow Ruoslahti's lead to reduce other integrin ligands to short peptides, including failures with laminin. That was a long time ago and really does not need to be an issue here.

Several important issues must be addressed to enhance the impact of the structural findings.

One of the drawbacks to this study is that the structure reveals a 'canonical' mode of RGD recognition with a correspondingly conventional change in conformation. This is not the author's fault, as the results speak for themselves and it was an important research question to address. However, the authors' discussion of the conceptual impact of their findings is not convincing. They state several times that the 'global conformational change' hypothesis for laminin/integrin complexes has been challenged in the literature. As evidence, they cite only 2 papers, citation 22, - a survey of complexes under various conditions using negative-stain EM with no other assays (J Cell Sci 2018), and two papers in citation 12. This appears to be a mistake; only the Takizawa paper (2012) is relevant. It concerns the laminin 511 crystal structure and mutagenesis that concluded with a model that the laminin gamma-chain contributes to both binding the $\alpha 6$ and $\beta 1$ subunit (which is also one of the main conclusions here). Furthermore, both of these papers are authored by the senior author of the current paper. Thus, this is hardly a controversy in the field; it is one between the same authors' own papers. They should not get credit for correcting their own previous conclusions, and instead should learn to be more cautious with their current conclusions, one of which comes in for significant criticism below. To amplify this point further, it is unclear why the senior author would have shown an EM micrograph (Fig. 3a in citation 12) that appears to show an open integrin $\alpha 6\beta 1$ conformation back in 2012 and then would have suggested in 2018 that $\alpha 6\beta 1$ did not open. Part of the mistake appears to stem from relying on the concept that the element Mn must always induce the high affinity state, as implied in line 83 of the MS where the "high affinity condition" actually means in the presence of Mn. Integrins are complicated. Although ligand binding almost always favors head opening for RGD-binding integrins, Mn does not always do so for RGD-binding integrins, as clearly illustrated with $\alpha v\beta 8$. Furthermore, there are exceptions even for ligands- there is no evidence that ligand binding favors head opening for $\alpha 4\beta 7$ or $\alpha v\beta 8$.

A second drawback is that the laminin/integrin structure does not appear to be exploited to gain insight into the mechanism by which the novel and extensive interactions regulate integrin activation. Rudimentary mutagenesis and binding studies have been performed to validate the interactions in the cryo-EM complex. However, there are no detailed functional or mechanistic studies, which would greatly increase the impact of the findings.

These and other issues are outlined in specific comments below:

Point 1, Lines 130-146 -The authors found that ADMIDAS is not occupied and suggested that TS2/16 activates $\beta 1$ integrin by expelling Ca^{2+} from ADMIDAS. This is a very dangerous conclusion. The authors' only other previous integrin crystal structure, of integrin $\alpha 5\beta 1$ bound to RGD, also was reported to lack Ca^{2+} at the ADMIDAS. This paper (Ref. 14) states "In addition to the change in MIDAS configuration, there was another critical change induced by RGD binding; the electron density corresponding to the ADMIDAS Ca^{2+} was diminished in the RGD-soaked crystal (Figs. 2 B and S3 and Video 3), whereas that of LIMBS, MIDAS, and all other metals bound to the $\alpha 5$ subunit remained unchanged. This ADMIDAS specific Ca^{2+} discharge seems inevitable because the two betaA residues that shifted most upon RGD binding, S134 and A342 (Fig. 4 A), provided their backbone carbonyl to ADMIDAS coordination." The interpretation here that Ca^{2+} discharge is inevitable is incorrect, because

many other RGD-binding integrins show complete Ca²⁺ retention when these same residues shift even more. Moreover, multiple subsequent RGD-liganded crystal structures of α5β1 showed complete Ca²⁺ retention, and the possibility that Ca²⁺ had been lost owing to its absence during soaking in ref. 14 was specifically raised.

The group submitting the current MS (and Ref. 14) appears to pay little attention to the very important issue of how crystal soaking is done, which is of extreme importance for proteins like integrins that bind metals with low affinity. Indeed, this lack of attention to important detail is especially notable in the current MS. The authors collected data at 100°K and almost certainly, given their precipitant (23% PEG1000, 0.2 M NaCl, 0.1 M Na/K phosphate pH 6.5) must have cryo-soaked. However, no description that cryoprotection was even done is provided, let alone whether Ca²⁺ was added to well solution. The authors must describe all the details, including an estimate of how long soaking was for, how many soaking steps were present, and whether Mg and Ca were added to the soaking solution. And the timing of soaking is almost never noted in the literature, but certainly has an effect; to give any useful estimate, the author who did this would have to repeat it with another crystal while someone else runs a stopwatch.

BTW, this is one example of how crucial data is omitted from the paper required to reproduce the results, in contrast to the overabundance of information that is not necessary on what failed to work.

Another factor is that the crystallization buffer contained 0.1 M Na/K phosphate and was below pH 7- at pH 6.5. Calcium and phosphate when mixed will bind to one another and can precipitate - they form bone and cements used in dental implants. Precipitation may not have occurred under these conditions; however, at a minimum, chelation of Ca²⁺ by phosphate will have lowered the concentration of free Ca²⁺.

Ca²⁺ is frequently found to be absent at the ADMIDAS in integrin crystal structures, including the first one, of αVβ3. Among the 6 or 7 metal-binding sites in the integrin head, the ADMIDAS is the one most sensitive to loss by low pH or soaking. This is understandable from first principles; the ADMIDAS has the fewest coordinating sidechains and backbone groups - only four- and no coordinating waters held in place by other residues

The artifacts of pH 6.5, Ca²⁺ chelation by phosphate, and soaking with a Ca²⁺ chelating buffer in the absence of added Ca²⁺, are more likely to describe the lack of Ca²⁺ at the ADMIDAS than the explanation on lines 145-146 that "TS2/16 may exert its activating effect through expelling ADMIDAS Ca²⁺ via unknown allosteric mechanism." The result section in 130-146 has some contradictions. First it is stated that "no change was induced in the α2 helix it contacts, precluding us from speculating on the allosteric mechanism." Two sentences later, they do speculate it is related to a "∼1.7Å move of the A342 carbonyl" but they do not say relative to what it moves. And then, as described above, they finally speculate that it works by expelling Ca²⁺. Remarkably, in each of their two integrin crystal structures, the group has found a loss of Ca²⁺ at the ADMIDAS but they are under different conditions. For α5β1 it was ascribed to partial movement toward the active state induced by ligand binding. For α6β1 there is no ligand bound when Ca²⁺ is "expelled." Instead, expulsion is attributed to a mysterious allosteric effect of TS2/16 binding. BTW, "discharged" (α5β1 MS) or "expelled" (α6β1) are much more colorful terms than the more common "lost" or "absent" appellations. Assuming that the authors are not attempting to bring attention to these effects by using colorful language, it would nonetheless be better not to imply that these are active processes. They are not, because the structures are at equilibrium, and metal binding will be determined by equilibria, even if their interpretations are correct.

Point 2. Surprisingly, the authors overlooked to analyze their own cryoEM structure and compare betaI (betaA) domain conformation and metal ion binding when TS2/16 is bound to α6β1 in the open conformation (cryoEM) and when TS2/16 is bound to α6β1 in the closed conformation (crystal structure). This comparison, facilitated by sharing of coordinate files with the reviewer, shows in the

open conformation relative the closed conformation, that within the TS2/16 binding site (epitope), the α 2-helix moves relative to other portions of the epitope including the neighboring β -strands, that there is a significant rearrangement of the Fab Tyr56 sidechain, and a possible increase in the size of the Fab footprint in the open conformation that should be checked with the PISA EBI server. The change in shape of the TS2/16 paratope provides a clear mechanism by which the Fab could favor the open conformation, if it is verified that TS2/16 binds to α 6 β 1 with higher affinity when α 6 β 1 is stabilized in the open conformation (when laminin or HUTS4 are present at saturating conditions), than when α 6 β 1 is stabilized in the closed conformation (SG/19?). The difference in affinity between the Fab for the closed and open conformations could be converted to the energy difference by which the Fab stabilizes the open over the closed conformation, and how TS2/16 works could finally be understood.

In passing, the binding of Mn to the ADMIDAS in the cryoEM structure further argues against the Ca expulsion model. And BTW, ref. 34 is misinterpreted. TS2/16 was seen bound to both the open and closed conformation in that paper.

Point 3. Lines 148-180 - much of this section deals with construct optimization, purification, and failed crystallization trials of the complex. The 'tail' seen in gel filtrations (Fig 2a, green peak) is not visually evident and peak symmetry or asymmetry looks similar upon addition of HUTS4. To make the suggestions on lines 157-162, Authors should perform SEC-MALS to see if the size of complexes changes along the peak, as evidence for dissociation. My suggestion is to condense this whole section and possibly put into supplementary data, since as it stands, it does not contribute significantly to the key findings of the paper. For the gel filtration profiles, if the authors subject the peak fractions to negative stain and can characterize the nature of the complexes, it would be a useful addition to the paper.

Point 4. Their complex structure revealed that, unlike the single-polypeptide RGD-type ligands, the three-chain laminin bears multiple integrin-binding determinants distributed in all chains, each contributing critically to the overall full binding activity. However, in their structure guided mutagenesis part, mutating each single residue D153, R155, R157 or N189 on α 6 submit almost completely abolished the interaction with laminin (Fig.5b). Also mutating K3099 on laminin completely abolished the interaction with integrin (Fig. 5c). Given the large interface between α 6 β 1 and LM511 (please quantitate buried SASA), it is surprising that a single mutation on either side would completely abolish binding. The charge-reversing mutations might cause other disruptive effects in integrin or laminin, such as altering sidechain conformation, or introducing steric overlap, rather than removing the specific interactions the study intended to examine.

More conservative mutations (Ala is the accepted one) would be much more useful for determining the contribution to binding energetics for each residue. The binding assays should be done in more quantitative ways to measure the effects of mutations on binding affinity, not merely a pull-down assay. The latter is qualitative, and has a very limited dynamic range, as clearly indicated by complete loss of binding with all but one mutation. Functional assays such as cell adhesion assays with different laminin coating concentration would also be informative. Much better would be quantitative measurements of affinity of a fluorescent laminin 511 E8 fragment with the integrin, which could be done by multiple methods either using purified material or flow cytometry without washing with α 6 β 1 transfectants. The latter could also be done using equilibrium binding to laminin 511 mutants by measuring their concentration dependent induction of Fluorescent HUTS4 or TS2/16 Fab binding to α 6 β 1 transfectants or native cells in flow cytometry.

Point 5. The authors make it seem that current understanding of integrin conformational states is limited to RGD-binding integrins. This is not so. Careful studies on α 4 β 1 have demonstrated that it has both bent and extended, and closed and open head states. Studies on multiple β 2 integrins have done exactly the same thing. It is true that laminin-binding integrins appear in early evolving metazoans, as do laminins. But this does not require naming those integrins in *C. elegans*, or such large emphasis on RGD integrins just because they also evolved early. BTW, the RGD-bearing ligands have changed in

evolution, while the laminins have always been the the ligands for laminin-binding integrins. The authors seem to talk about RGD binding integrins and all of their hard work as the main story lines, but scientists get credit for their discoveries, not for their work.

Examples include, but are not limited to, lines 115-119: "Importantly, the two resolved $\alpha 6$ domains (β -propeller and thigh) provide the first atomic view of the integrin α subunit of the laminin binding class. As predicted from the sequence conservation pattern, the seven-bladed β -propeller domain of the $\alpha 6$ subunit bound three Ca^{2+} ions at the bottom of the blades V to VII that are conserved in all integrin subunits (Supplementary Fig. 1), while no Ca^{2+} was observed at the blade IV that is specific for RGD-binding integrins (Supplementary Fig. 1)."

Blade IV that lacks Ca was already observed in crystal structures of the β -propeller domains of $\alpha 4$, αL , and αX subunits and is nothing new.

Lines 120-122: The observations on what is conserved in beta-propeller domains has already been described many times.

Point 6. Instead of using the hard work and the stepwise progression from crystal to cryoEM as the story line, the paper would be much better written by focusing on structural features, and describing the closed, unliganded and open, liganded structures in parallel rather than in sequence. Thus, how TS2/16 binds would be much better addressed by comparing the two structures. Similarly, the unusual alternatively spliced regions, the subject of the next three points, would be much better described in parallel.

Point 6. The unique features of laminin binding integrins, the X or X1 and X2 regions in their β -propeller domains, which are alternatively spliced in some laminin-binding integrin α subunits, get only minor attention but are their most fascinating feature. They are described in lines 123-129, buried at the end of a paragraph with the topic sentence "To gain structural insights into the laminin-integrin interaction, we first aimed at crystallizing human $\alpha 6\beta 1$ integrin ectodomain." They are next described towards the end of Results, lines 254-264. However, their significance is also somewhat unclear, and only with careful reading is it clear that they form part of the ligand binding site! This most interesting feature is also not mentioned in the abstract, where the focus is still on multiple chains in laminin contributing to ligand recognition. One is led not to expect much from this article because the abstract concludes "...laminin recognition by $\alpha 6\beta 1$ integrin follows the "canonical" rule of integrin-ligand interaction, where it involves a direct coordination of a ligand carboxylate to the metal-ion dependent adhesion site of integrin $\beta 1$ subunit, leading to a large-scale shape-shifting of the receptor." (The large shape shifting was already described in an integrin crystal structure in 2004 for $\alpha \text{IIb}\beta 3$.) The main point a reader takes away from this MS is how the investigators are real pioneers, and have toiled relentlessly and creatively to complete this work. By contrast, the biological significance of alternative splicing of the ligand binding segment, unique to laminin-binding integrins, is not clearly brought out.

Point 7. Lines 123-125. This reviewer finds the most interesting sentence in the MS here: "We noted that the electron density is particularly weak for the segment spanning K199 to V227 with only the central β -sheet element 211DGPYEV216 showing traceable electron density (Fig. 1c), suggesting a mobile nature of this entire segment including the edge strand of blade III." Whoa, the implications of this are amazing. The Authors are saying that register of the sequence to the structure could shift on the edge strand. But in Fig. 1c it is hard to get the context and realize that this sequence is very close to the laminin binding site. And the text only hints at this possibility and asks the reader to wait until much later in the paper, and the possibility of the register shift is never brought up again. A view of this edge β -strand comes up again in Fig. 4c, but there is no comparison to Fig. 1c, which is distant in the reader's memory here, and the residues in the β -strand in Fig. 1c are shown in stick in Fig. 4c (This reviewer had to read the Figure legend carefully to figure out that this was the edge β -strand, since it was shown all in stick). It is even not made clear if the register is the same in the closed and open structures, and whether there is good experimental evidence for that. Furthermore in that part of the MS, the topic sentence is "Within the integrin-laminin interface, the $\alpha 6$ subunit provides more

contacts than the $\beta 1$, contributing toward $\sim 70\%$ of the total interface." BTW, please describe the total solvent-accessible surface buried on each side of this interface.

Much of the importance of the edge beta strand is lost because the Authors discuss the "I-linker loop" in the same paragraph and same figure. BTW, "I-linker loop" is unclear terminology, because its importance in ligand recognition is only in integrins that LACK betaI domains. And an I-linker loop is actually one of the two linkers that connect a b-propeller domain to the inserted I domain, not a loop in an integrin that lacks an I domain. It would be much better to use previous terminology such as the blade II-blade III loop, or more concise, the W2-W3 loop or W2 beta4-beta1 loop (W is an abbreviation for blade in the beta-propeller field, as each blade has a W-like topology). Furthermore, the importance of this same loop in ligand recognition has already been described in other integrins, so why change the name in this paper?

Fig. 4e could be greatly simplified- so many integrins are not needed. The positions of each blade and b-strand should be labeled. Abbreviations such as W3,beta4 could be used. Thus, the important position of W3,beta4, the edge beta-strand where register shifting might occur, could be labeled in the middle of the highlighted X1 region. Even the conventions used and coloring and the description of this figure in the legend need much improvement. The figure legend says the region shown is "near the boundary between blade II and III." This lacks important detail, because the region shown is actually blade III, and includes the boundaries with both blades II and IV. The authors should not use different colors to show residues mutated in this study or in Ref. 43 and instead should name the residues mutated in the legend. Furthermore, they should highlight residues mutated and found important in ligand binding, not simply "residues mutated."

Figure panels 4a-d also need much improvement. There must be a better way to display the laminin-integrin interaction site, including open-book views, possibly two types, one showing interacting residues as sticks with remainder as cartoon, and another as solvent accessible surface, with colors showing regions that are in contact. All easily done with pymol commands such as "select contact, br. A within 4.0 of B" for stick display and omitting "br." for surface display. Panel 4b is extremely difficult to see, to read labels or see what they label, and needs complete redoing.

Point 8. The authors should test the register-shifting hypothesis to bring the biological significance of this article up to Nature Communication standards. These are not difficult experiments. The methodology is straightforward, simple, and already described in Point 4. Highly quantitative measurements can be done of the affinity at equilibrium of laminin fragments to integrins on cells by measuring the concentration dependence of induction by ligand of fluorescent HUTS4 or TS2/16 Fab fragment binding to integrin transfectants using flow cytometry with no washing. As a hint, if the Authors need to increase laminin affinity to save on Fab, they can add an unlabeled extension-stabilizing Fab to the assay. Register shifting is easily achieved by truncating the 194-235 loop stepwise in the W3 beta3-4 loop to shift the W3 beta 4 strand to a more C-terminal sequence register or in the W3 beta4-1 loop to shift the register more N-terminal.

The Authors should test the hypothesis that the register can shift to allow laminin-binding integrins to bind to different laminins. The Authors report no register shift when $\alpha 6\beta 1$ binds laminin 511. Therefore, the lowest energy register fits laminin 511, consistent with $\alpha 6\beta 1$ having highest affinity for this laminin. The 194-235 loop contacts the laminin alpha subunit, coded as "5" in the alpha-beta-gamma numbering scheme (abc). The Authors should confirm that other laminins with different alpha subunits bind $\alpha 6\beta 1$ with lower affinity. Then they should test register-shifting deletions. The prediction is that register-shifting deletions would lower affinity for 511, by changing the low energy register position, and by changing the equilibrium toward more N or C-terminal register positions, increase or decrease the affinity for other laminins depending on whether their lowest affinity registers are toward the C or N-terminal positions of the 194-235 loop, respectively. Furthermore, the 194-235 loop should be substituted by the X1, X2, or X1X2 positions of the integrin $\alpha 3$ and $\alpha 7$ subunits (From the text, it was difficult to discern whether $\alpha 6$ has an X2 splice variant, if it does, this should of course also be tested).

The Authors fail to discuss the features of the alpha6 X1 loop that favor a particular register. Y214 and V216 fit into hydrophobic pockets of the b-propeller and are therefore key. The most likely alternative register is achieved when F204 and M206 fit into the same pocket.

If the Authors demonstrate register shifting, it would have enormous importance for the co-evolution of laminins and their cognate integrin receptors from the first metazoans. It would demonstrate a quite different pattern than for RGD integrins, which switched from recognizing one ligand, which disappeared in evolution, to recognizing newly evolving ligands, such as fibronectin. In contrast, laminin and laminin-recognizing integrins would be shown to have succeeded in co-evolving, owing to alternative splicing (X1 and X2) and register-switching in X1 and X2. Together, these two mechanisms would have provided flexibility for new laminin alpha subunits to evolve, without a loss in recognition by integrins, and with the flexibility to evolve new laminins to fulfill new functions in organisms.

This achievement would not only ensure acceptance by NatCommun, but likely acceptance in a sister journal such as Nature.

Point 9. There are multiple criticisms on the structural work for not describing it thoroughly and pushing it further. There is continuous density for the 194-235 loop and so it can be built, for example with PHENIX.ROSETTA which is very helpful at their 3.9 Å resolution. And would also likely greatly improve the fit for the rest of the model, and perhaps for regions of the crystal structure. They should push cryoEM further as described above and describe how they refined. They mention rigid body fitting of both Fv fragments, but obviously refined them as well. They removed HUTS4 Fv for part of the particle registration, but obviously used it in refinement, please describe. Their data cutoffs in Supplementary Table 1 are far too old-fashioned (conservative); they are throwing away much useful data, and need to read and learn modern data cutoff best practices in the paper by Karplus and Diederichs. They fail to show R_{work}/R_{free} in the highest resolution shell. No results on MolProbity are shown, the use of which in model building (NOT in refinement) could greatly improve their models.

In summary, this intriguing structural report seems 'under-exploited' with respect to its impact on integrin recognition by laminins. The authors are urged to perform functional/cellular studies to address key questions that remain unanswered.

Minor issues

-The figures need substantial revisions. Colors are inconsistent - for example, in 1b, the salmon colored domains are 'raspberry' color in 1c. In line 177 of paper, fig 1a is cited, but the text refers to EM images in the literature. The antibody in Fig 1d (cartoon transparency) lies on top of Ca vectors. Also, the colors of TS2/16 antibody (green/blue) clash with b1 subunit (blue). Fig 1e could be greatly simplified - the only difference is the small movement of the a-helix.

- line 77 - fig 1a citation unusual - refers to EM images

- line 99 - replace 'warranted by an aggregation...' by 'mediated by a cluster of small and potentially weak interactions that come from residues distributed across all five subunits'

- 113-114 - not clear which parts of b1 domain not resolved - is PSI domain at both N/C termini? Figure needs to clarify this

- line 124 - difficult to see in Fig1c what they are referring to (residues 211-216)

- line 126 - change 'strange' to 'unusual'. In the current figures, it is difficult to see which parts of b-propeller are unusually mobile, perhaps an additional figure here or in suppl data would help?

- 130-132, very long, run-on sentence, please break and simplify.
- 144-145, 'and is suggested to be...'
- 146 'an unknown'
- line 327, no contractions (couldn't)

Reviewer #3 (Remarks to the Author):

This study provided by Takagi and colleagues reports a cryo-EM and crystallographic structural analyses of $\alpha 6\beta 1$ integrin alone and in complex with laminin.

Integrins are important cell surface receptors that connect to proteins in the extracellular matrix like laminin and mediate cell adhesion. The presented work revealed how laminin subunits interact with the pocket between the α - and β -integrin heads using a set of conformation-fixing antibodies. Interestingly, using laminin LM511 fragment E8 provided quite a different picture of the interaction compared to published RGD-based ligands or ligand mimetics. It showed a complex binding interface with multiple contact points, and it highlights the uniqueness of the $\alpha 6\beta 1$ -laminin interaction. Key residues are gamma1E1607 in LM511, which directly coordinates the MIDAS metal in integrin $\beta 1$, and P1609, which contacts the conserved N189 of $\alpha 6$ integrin, in an analogous way as the conventional RGD-based ligand.

This study provides a number of highly relevant residues on laminin and integrin subunits that are key for the fixation of the integrin head opening and therefore, I fully support this study to be published at Nature Communications.

One comment:

In the cryo-EM map, do you actually see the ADMIDAS metal ion? Fig. 4 implies that the metal is there. The authors should clarify this point, as they did not see the ADMIDAS metal ion in the closed integrin without ligand.

Response to the reviewers' comments

RE: NCOMMS-20-37858-T

Arimori et al.

" Structural mechanism of laminin recognition by integrin"

Followings are our point-by-point responses to the comments and requests provided in the decision letter we received on Nov 3, 2020. Requests/comments are in *black and italicized* and our responses are in **red**.

Comments by the reviewers

Reviewer #1:

Remarks to the Author:

This is an outstanding piece of work that finally describes the structural basis for the binding of the binding of integrin alpha6 beta 1 to laminin 511. The authors employed both crystallography and cryo EM to solve the structures. The work is technically outstanding and there are no major criticisms of the paper. It has solved a longstanding problem in the field of integrin biology and is therefore highly significant to all of us who work in this field.

We thank the reviewer very much for highly valuing our work.

There are a few minor comments that would help to improve the paper.

*1) While the English is very good there are some grammatical errors that should be fixed prior to publication.
2) In line 138 the authors state "Interestingly, this shape-shifting is accompanied by a side-chain flipping of the D137, which, together with a ~1.7 Å move of the A342 carbonyl, results in the disruption of the coordination environment of the ADjacent to MIDAS (ADMIDAS) metal (Fig. 1e). It would be nice to show a supplemental figure of the density map and how the relevant sidechains fit as their resolution is only 2.5 to 2.9 Å.*

We revised the entire MS and have gone over the text to eliminate grammatical errors as much as possible. Regarding the point 2), we added a density map near this region as the Supplementary Fig 1b. Due to the low resolution, it is difficult to tell if the map strongly supports the sidechain orientation of D137, but at least we can safely say that it is incompatible with the ADMIDAS-coordinating conformation.

2)Space permitting by the editors it would be nice to show supplemental figure 4 in the main body as it is important and interesting data.

We thank the reviewer for this kind suggestion. As we reduced the total number of our main figure from 6 to 5, we could theoretically move some of the former Supplementary Fig.4 subpanels (e.g., the EM density map in Supplementary Fig.4c in the revised MS) to the main body of the MS without taking much space away. We would like to defer to the editor's decision about this point.

3) Panel A in supplemental figure shows very dense particles. It would be nice to see one where the particles are more dilute.

We assume that the reviewer's concern is that there may be too much particle overlaps. The field view looks very dense because of the high contrast, thanks to the use of phase plate during the data acquisition. In fact, most particles are well separated. We added an expanded view as an inset to the Supplementary Fig.3a to show this point.

4) There appears to be an error in line 77 where figure 1A is referenced. This should be fixed

The reason we cited this figure here was to show that the top quadrangle of the scheme (laminin binding class) does not contain structurally determined integrins (circled by thick black line). We realized that this reference to Fig.1a is confusing because it also confused reviewer #2. So we added explanation in the parenthesis in line 86.

Reviewer #2:

Remarks to the Author:

This paper describes the structure of the laminin-integrin $\alpha 6 \beta 1$ complex by cryo-EM and crystallography. Although numerous integrin complexes have previously been determined, there is no structural information on how laminins bind integrins. The work described includes construct optimization, conformation-stabilizing antibodies, X-ray structures of components including the conformation-stabilizing Fv-clasp construct of antibody HUTS4 and a complex of the integrin $\alpha 6 \beta 1$ headpiece bound to TS2/16 Fv-clasp. These components were then assembled with a fragment of laminin (LM511) to form a high-affinity complex that was suitable for cryo-EM structure determination. This heroic effort was required to make a landmark achievement, the first molecular view of how integrins recognize laminin.

We thank the reviewer for the careful reading and highly valuing our work.

However, the impact of this achievement is offset by many distractions. These include the way the paper is written, how the subject is introduced, too much emphasis on all the dead ends and hard work, much description of what did not work, tendentious claims including controversy on integrin head opening and how integrins work, over-emphasis on RGD-binding integrins, which ignores important achievements to date on how non-RGD binding integrins work, and a few doubtful interpretations that are really not necessary to the main thrust of the paper. Furthermore, the figures poorly illustrate the main point of how laminin is bound and do not clearly illustrate the overall binding site.

The overall work overcomes numerous obstacles to previous efforts on this complex, although no specific approach by itself is novel. The cryo-EM structure is intriguing and reveals a major footprint of LM511 on the $\alpha 6$ integrin subunit, but its size is not quantitated, the contacts could be put into better context, and their relative importance is not measured quantitatively. One wonders why the EM resolution was only 3.9Å, whereas the group of Yifan Chen was able to achieve substantially higher resolution on the complex of the entire $\alpha V \beta 8$ ectodomain bound to proTGF- $\beta 1$, an even larger complex.

The $\alpha V \beta 8$ structure, however, focused refinement only on the center of the complex. Whether this was done here is not defined. The paper would be greatly improved if the EM was at higher resolution and details were better described.

The authors need to put much more work into carefully interpreting the structure, their conclusions and writing about what is novel in their structure. And spend less time on points that are peripheral to their point on laminin. The previous laminin crystal structures already made clear that all three chains come closely together at the key Asp sidechain that is recognized, and must all be required for integrin recognition, so there is really no need to stress the distinction from RGD. Yes, lots of other people tried to follow Ruoslahti's lead to reduce other integrin ligands to short peptides, including failures with laminin. That was a long time ago and really does not need to be an issue here.

We regret that the way we wrote the paper was judged inappropriate/insufficient in conveying the scientific value of the work by this reviewer. We took these criticisms seriously, and revised the manuscript by re-writing the large body of the text focusing more on the actual data and their interpretations, re-analyzing the structure to validate our claims, and adding quantitative data by additional new experiments, which will be described in detail in the following sections.

Several important issues must be addressed to enhance the impact of the structural findings.

One of the drawbacks to this study is that the structure reveals a ‘canonical’ mode of RGD recognition with a correspondingly conventional change in conformation. This is not the author’s fault, as the results speak for themselves and it was an important research question to address. However, the authors’ discussion of the conceptual impact of their findings is not convincing. They state several times that the ‘global conformational change’ hypothesis for laminin/integrin complexes has been challenged in the literature. As evidence, they cite only 2 papers, citation 22, - a survey of complexes under various conditions using negative-stain EM with no other assays (J Cell Sci 2018), and two papers in citation 12. This appears to be a mistake; only the Takizawa paper (2012) is relevant. It concerns the laminin 511 crystal structure and mutagenesis that concluded with a model that the laminin gamma-chain contributes to both binding the $\alpha 6$ and $\beta 1$ subunit (which is also one of the main conclusions here). Furthermore, both of these papers are authored by the senior author of the current paper. Thus, this is hardly a controversy in the field; it is one between the same authors' own papers. They should not get credit for correcting their own previous conclusions, and instead should learn to be more cautious with their current conclusions, one of which comes in for significant criticism below. To amplify this point further, it is unclear why the senior author would have shown an EM micrograph (Fig. 3a in citation 12) that appears to show an open integrin $\alpha 6\beta 1$ conformation back in 2012 and then would have suggested in 2018 that $\alpha 6\beta 1$ did not open. Part of the mistake appears to stem from relying on the concept that the element Mn must always induce the high affinity state, as implied in line 83 of the MS where the "high affinity condition" actually means in the presence of Mn. Integrins are complicated. Although ligand binding almost always favors head opening for RGD-binding integrins, Mn does not always do so for RGD-binding integrins, as clearly illustrated with $\alpha v\beta 8$. Furthermore, there are exceptions even for ligands- there is no evidence that ligand binding favors head opening for $\alpha 4\beta 7$ or $\alpha V\beta 8$.

This big criticism is based on a simple misunderstanding about what we were trying to say, and it is our fault to cause this misunderstanding by some ambiguous and ill-defined terminologies. First of all, by "global conformational change hypothesis" (line 81; by the way, we stated this only once here) we meant an idea that integrin global conformational change (=extension) is tightly linked with the acquisition of high ligand affinity. That's why we cited ref 21 (Miyazaki, JCS 2018) which showed that there was no essential difference in the global shape of $\alpha 6\beta 1$ integrin before and after activation by Mn^{2+} . Because of the beautiful bent-extend conversion images on $\alpha v\beta 3$ and $\alpha L\beta 2$ integrins published repeatedly from Tim Springer lab, this misconception (i.e., integrin activation must accompany large conformational change) is still prevalent among, at least, non-experts. In contrast, the linkage between ligand binding and integrin extension (or at least the head opening) is already well established before the current work, as the reviewer points out, and we agree that we have just confirmed this for laminin-binding integrin at higher resolution than before. So there is no contradictions between Takizawa 2017 (not 2012) paper and Miyazaki 2018 paper because the latter did not image laminin-bound integrins. That being said, we realized that the use of this terminology itself may create further confusions, because it confused an expert like this reviewer already. So we decided to change these parts (lines 80-90 in Introduction and lines 375-383 in the Discussion) to simply introduce the status of integrin structural analyses in relation to the ligand binding, and to point out the fact that high resolution structure for laminin-integrin interaction was missing.

A second drawback is that the laminin/integrin structure does not appear to be exploited to gain insight into the mechanism by which the novel and extensive interactions regulate integrin activation. Rudimentary mutagenesis and binding studies have been performed to validate the interactions in the cryo-EM complex. However, there are no detailed functional or mechanistic studies, which would greatly increase the impact of the findings.

These and other issues are outlined in specific comments below:

We thank the reviewer for numerous constructive comments with valuable suggestions. We provide our answers to each of them point-by-point below.

Point 1, Lines 130-146 -The authors found that ADMIDAS is not occupied and suggested that TS2/16 activates $\beta 1$ integrin by expelling Ca^{2+} from ADMIDAS. This is a very dangerous conclusion. The authors' only other previous integrin crystal structure, of integrin $\alpha 5\beta 1$ bound to RGD, also was reported to lack Ca^{2+} at the ADMIDAS. This paper (Ref. 14) states "In addition to the change in MIDAS configuration, there was another critical change induced by RGD binding; the electron density corresponding to the ADMIDAS Ca^{2+} was diminished in the RGD-soaked crystal (Figs. 2 B and S3 and Video 3), whereas that of LIMBS, MIDAS, and all other metals bound to the $\alpha 5$ subunit remained unchanged. This ADMIDAS specific Ca^{2+} discharge seems inevitable because the two βA residues that shifted most upon RGD binding, S134 and A342 (Fig. 4 A), provided their backbone carbonyl to ADMIDAS coordination." The interpretation here that Ca^{2+} discharge is inevitable is incorrect, because many other RGD-binding integrins show complete Ca^{2+} retention when these same residues shift even more. Moreover, multiple subsequent RGD-liganded crystal structures of $\alpha 5\beta 1$ showed complete Ca^{2+} retention, and the possibility that Ca^{2+} had been lost owing to its absence during soaking in ref. 14 was specifically raised.

The reviewer cite our statement in ref 14 (the first crystal structure of $\beta 1$ integrin), which may have misinterpreted the nature of the ADMIDAS metal behavior because later higher resolution crystal structures from Springer lab retained Ca^{2+} . We cannot take out those statements in our published paper. We agree that the choice of word "expel" in the current MS was not appropriate and changed the sentence to avoid potentially misleading description about the observed data. (see also the next two answers)

The group submitting the current MS (and Ref. 14) appears to pay little attention to the very important issue of how crystal soaking is done, which is of extreme importance for proteins like integrins that bind metals with low affinity. Indeed, this lack of attention to important detail is especially notable in the current MS. The authors collected data at 100°K and almost certainly, given their precipitant (23% PEG1000, 0.2 M NaCl, 0.1 M Na/K phosphate pH 6.5) must have cryo-soaked. However, no description that cryoprotection was even done is provided, let alone whether Ca^{2+} was added to well solution. The authors must describe all the details, including an estimate of how long soaking was for, how many soaking steps were present, and whether Mg and Ca were added to the soaking solution. And the timing of soaking is almost never noted in the literature, but certainly has an effect; to give any useful estimate, the author who did this would have to repeat it with another crystal while someone else runs a stopwatch.

BTW, this is one example of how crucial data is omitted from the paper required to reproduce the results, in contrast to the overabundance of information that is not necessary on what failed to work.

Another factor is that the crystallization buffer contained 0.1 M Na/K phosphate and was below pH 7- at pH 6.5. Calcium and phosphate when mixed will bind to one another and can precipitate - they form bone and cements used in dental implants. Precipitation may not have occurred under these conditions; however, at a minimum, chelation of Ca^{2+} by phosphate will have lowered the concentration of free Ca^{2+} .

Ca^{2+} is frequently found to be absent at the ADMIDAS in integrin crystal structures, including the first one, of $\alpha V\beta 3$. Among the 6 or 7 metal-binding sites in the integrin head, the ADMIDAS is the one most sensitive to loss by low pH or soaking. This is understandable from first principles; the ADMIDAS has the fewest coordinating sidechains and backbone groups - only four- and no coordinating waters held in place by other residues

We apologize that we did not provide enough information about the cryoprotection conditions in the MS. Now we revised the method section to include this information. For the $\alpha 6\beta 1$ -TS2/16 complex crystal, we actually did not perform soaking at all; the crystals obtained in the indicated buffer condition were directly flash-frozen in liquid nitrogen without cryo-soaking. We strongly agree that experimental details should be provided to enable others to reproduce the result, but in this case it just did not occur to us that it was important to describe the experimental steps we did not perform. Omission of cryo-soaking was possible probably because the presence of 23% PEG functioned as cryoprotectant. We always test if additional cryoprotection is essential or not, and when possible, choose direct freezing to avoid potential dilution of the buffer constituents. As the crystallization drop was made by mixing sample solution (in TBS containing 1mM Ca and Mg) with the crystallization buffer (23% PEG1000, 0.2 M NaCl, 0.1 M Na/K phosphate pH 6.5) at 1:1 ratio, the nominal final concentration of Ca^{2+} was 0.5 mM or may be slightly higher due to the evaporation, and no

dilution had occurred as described above. This condition is very similar to what employed in the crystallization of non-liganded $\alpha 5b1$ headpiece by Springer and colleagues (4wjk, 1.85Å), where ADMIDAS Ca^{2+} is clearly present with an expected occupancy of >90%. As the reviewer pointed out, however, there was a critical difference between the two conditions, because the crystallization buffer of 4wjk was 0.1M Hepes, pH7.2. Presence of 0.1M phosphate in our condition will indeed lower the effective concentration of free Ca^{2+} and this may have contributed to the preferential loss of ADMIDAS Ca^{2+} , irrespective of the TS2/16 binding state. We have revised the MS to incorporate this possibility (lines 156-159) and completely eliminated our speculation about the loss of Ca^{2+} as the potential mechanism of activation by TS2/16.

The artifacts of pH 6.5, Ca^{2+} chelation by phosphate, and soaking with a Ca^{2+} chelating buffer in the absence of added Ca^{2+} , are more likely to describe the lack of Ca^{2+} at the ADMIDAS than the explanation on lines 145-146 that "TS2/16 may exert its activating effect through expelling ADMIDAS Ca^{2+} via unknown allosteric mechanism." The result section in 130-146 has some contradictions. First it is stated that "no change was induced in the $\alpha 2$ helix it contacts, precluding us from speculating on the allosteric mechanism." Two sentences later, they do speculate it is related to a " $\sim 1.7\text{\AA}$ move of the A342 carbonyl" but they do not say relative to what it moves. And then, as described above, they finally speculate that it works by expelling Ca^{2+} . Remarkably, in each of their two integrin crystal structures, the group has found a loss of Ca^{2+} at the ADMIDAS but they are under different conditions. For $\alpha 5b1$ it was ascribed to partial movement toward the active state induced by ligand binding. For $\alpha 6b1$ there is no ligand bound when Ca^{2+} is "expelled." Instead, expulsion is attributed to a mysterious allosteric effect of TS2/16 binding. BTW, "discharged" ($\alpha 5b1$ MS) or "expelled" ($\alpha 6b1$) are much more colorful terms than the more common "lost" or "absent" appellations. Assuming that the authors are not attempting to bring attention to these effects by using colorful language, it would nonetheless be better not to imply that these are active processes. They are not, because the structures are at equilibrium, and metal binding will be determined by equilibria, even if their interpretations are correct.

Again, we agree that we lack a clear evidence of connecting the activation mechanism of TS2/16 and the absence of Ca^{2+} , so removed the entire segment and just stated about the lack of Ca^{2+} as a fact. We did not intend to imply that this was an active process by using a word "expel" (or "discharge" in ref 14). If it sounded that way, that was our fault and we apologize for our lack of English proficiency. Regarding the activation mechanism by TS2/16, please see our answer to the Point 2.

Point 2. Surprisingly, the authors overlooked to analyze their own cryoEM structure and compare beta1 (betaA) domain conformation and metal ion binding when TS2/16 is bound to $\alpha 6b1$ in the open conformation (cryoEM) and when TS2/16 is bound to $\alpha 6b1$ in the closed conformation (crystal structure). This comparison, facilitated by sharing of coordinate files with the reviewer, shows in the open conformation relative the closed conformation, that within the TS2/16 binding site (epitope), the $\alpha 2$ -helix moves relative to other portions of the epitope including the neighboring b -strands, that there is a significant rearrangement of the Fab Tyr56 sidechain, and a possible increase in the size of the Fab footprint in the open conformation that should be checked with the PISA EBI server. The change in shape of the TS2/16 paratope provides a clear mechanism by which the Fab could favor the open conformation, if it is verified that TS2/16 binds to $\alpha 6b1$ with higher affinity when $\alpha 6b1$ is stabilized in the open conformation (when laminin or HUTS4 are present at saturating conditions), than when $\alpha 6b1$ is stabilized in the closed conformation (SG/19?). The difference in affinity between the Fab for the closed and open conformations could be converted to the energy difference by which the Fab stabilizes the open over the closed conformation, and how TS2/16 works could finally be understood. In passing, the binding of Mn to the ADMIDAS in the cryoEM structure further argues against the Ca expulsion model. And BTW, ref. 34 is misinterpreted. TS2/16 was seen bound to both the open and closed conformation in that paper.

First of all, we have to apologize that the Tyr56 sidechain of TS2/16 H chain was wrongly modeled in the provided EM structure. The EM map suggests that the orientation is the same as the crystal structure (see image below) so we amended the coordinates. It must have accidentally flipped during the refinement process. We did compare the binding interface of TS2/16 on betaA domain between the two structures but did

not mention it in the MS, because the structural difference at betaA domain (after the superposition using the Fv region) was negligible when we consider the resolution of the structures. We added a new figure showing this point in the revised MS (Supplementary Fig.5c), and corrected the misinterpretation of the ref. 34 (line 146). Using the updated coordinates, we calculated the interface sizes using the PISA EBI server as suggested, and obtained following values; X-ray(closed): b1-VH: 352.7 Å², b1-VL: 533.6 Å², EM(open): b1-VH: 341.1 Å², b1-VL: 522.6 Å². As the footprint area for the open betaA was not larger than the closed form, we cannot say that TS2/16 favors open conformation. However, we agree with the reviewer that the activation mechanism by TS2/16 can be energetically explained, if we can prove that TS2/16 binds stronger to the open form than the closed one. So we decided to perform the following experiment. First, we constructed fluorescent TS2/16 by fusing sfGFP at the C-terminal of the heavy chain of TS2/16 Fv-clasp. Then we evaluated the binding of TS2/16-sfGFP to K562 cells (which express α5β1 as the sole beta1 integrin) in the resting state (i.e., in 1mM Ca²⁺ & Mg²⁺, where most integrins are expected to assume close conformation) or in the activated state (in 0.1mM Ca²⁺, 1 mM Mn²⁺ plus 1mM RGD peptide, where open conformation should be predominant). We could not perform similar experiments for α6β1 because most cell lines express several different alpha subunits that can couple with β1 subunit, complicating the interpretation of the TS2/16 binding data. As shown in the new Supplementary Fig.5d, the binding isotherm was not affected significantly by the conformational state, leading to a K_d value of ~1.3 nM under both conditions. Therefore, despite all the valuable comments and suggestion by the reviewer and our efforts to address the issue accordingly, we still do not understand how TS2/16 activates beta1 integrin. Nevertheless, these data and discussions are included in the revised MS (lines 230-237). I hope this information can help others to solve the mystery in the future.

Point 3. Lines 148-180 - much of this section deals with construct optimization, purification, and failed crystallization trials of the complex. The 'tail' seen in gel filtrations (Fig 2a, green peak) is not visually evident and peak symmetry or asymmetry looks similar upon addition of HUTS4. To make the suggestions on lines 157-162, Authors should perform SEC-MALS to see if the size of complexes changes along the peak, as evidence for dissociation. My suggestion is to condense this whole section and possibly put into supplementary data, since as it stands, it does not contribute significantly to the key findings of the paper. For

the gel filtration profiles, if the authors subject the peak fractions to negative stain and can characterize the nature of the complexes, it would be a useful addition to the paper.

The purpose of having this section was to show the systematic comparison among various a6b1 complex species in their stability under different conditions, rather than just introducing the final condition of sample prep that were successful. Nevertheless we agree that much of the description is not essential, so we have moved this entire section and the Fig.2 to supplementary section. As to the peak symmetry, addition of the HUTS-4 was not intended to stabilize the complex (i.e., make the peak more symmetrical), but to prevent spontaneous closure of the swung-out hybrid domain by acting like a wedge. This point is also added in the revised MS (lines 166-168).

Point 4. Their complex structure revealed that, unlike the single-polypeptide RGD-type ligands, the three-chain laminin bears multiple integrin-binding determinants distributed in all chains, each contributing critically to the overall full binding activity. However, in their structure guided mutagenesis part, mutating each single residue D153, R155, R157 or N189 on alpha 6 submit almost completely abolished the interaction with laminin (Fig.5b). Also mutating K3099 on laminin completely abolished the interaction with integrin (Fig. 5c). Given the large interface between a6b1 and LM511 (please quantitate buried SASA), it is surprising that a single mutation on either side would completely abolish binding. The charge-reversing mutations might cause other disruptive effects in integrin or laminin, such as altering sidechain conformation, or introducing steric overlap, rather than removing the specific interactions the study intended to examine.

More conservative mutations (Ala is the accepted one) would be much more useful for determining the contribution to binding energetics for each residue. The binding assays should be done in more quantitative ways to measure the effects of mutations on binding affinity, not merely a pull-down assay. The latter is qualitative, and has a very limited dynamic range, as clearly indicated by complete loss of binding with all but one mutation. Functional assays such as cell adhesion assays with different laminin coating concentration would also be informative. Much better would be quantitative measurements of affinity of a fluorescent laminin 511 E8 fragment with the integrin, which could be done by multiple methods either using purified material or flow cytometry without washing with a6b1 transfectants. The latter could also be done using equilibrium binding to laminin 511 mutants by measuring their concentration dependent induction of Fluorescent HUTS4 or TS2/16 Fab binding to a6b1 transfectants or native cells in flow cytometry.

We thank the reviewer for these valuable suggestions. We agree that the pull-down assay tends to give yes-no answers and is not ideal to evaluate contribution of each mutation to the binding which are often partial. So we decided to perform more quantitative analyses on the laminin-integrin interaction by using fluorescently labeled laminin E8 fragments. To this end, we fused sfGFP to the N-terminal of truncated laminin alpha5 subunit and co-expressed with truncated beta1 and gamma1. The resultant sfGFP-tLM511E8 was purified and incubated with a6b1-expressing cells at defined concentrations and the binding was evaluated by FACS without washing, as suggested. Preliminary experiments with WT a6b1-transfected cells indicated that the binding was rather weak under the resting condition (i.e., 1mM Ca & Mg), precluding us from deriving meaningful Kd value. However, in the presence of 0.5mM Mn, we could see concentration-dependent E8 binding in an a6b1-dependent manner (See Supple Fig.6a), and the plot of the integrated fluorescence values (in a linear scale, rather than the popular log-scale MFI values) with the free E8 concentrations derived Kd values of ~13 nM for interaction between WT E8 and WT a6b1. By using this assay system, we have evaluated the laminin-binding activity of a more elaborate set of integrin mutants (14 compared to just 5 in the previous MS), particularly those with more conservative Ala mutations according to the suggestion. As a result, we indeed found that D153 and R157 contributed partially to the binding (causing 3.6- and 2.3-fold increase of Kd values, respectively), while R155 and N189 were absolutely required because Ala mutation abolished the binding. We also expanded our mutational search to residues in the X1 regions and found that acidic residues in this region contribute to the binding at variable levels. (discussed later in the answer to the Point 8)

As a result of these additional experiments and data, we expanded the relevant section in the main text (lines 306-326) as well as the Method section, and added new Fig.4d-g and Supplementary Fig 6. We also added buried SASA values for each side of the complex in page 8, line 243.

*Point 5. The authors make it seem that current understanding of integrin conformational states is limited to RGD-binding integrins. This is not so. Careful studies on $\alpha 4\beta 1$ have demonstrated that it has both bent and extended, and closed and open head states. Studies on multiple $\beta 2$ integrins have done exactly the same thing. It is true that laminin-binding integrins appear in early evolving metazoans, as do laminins. But this does not require naming those integrins in *C. elegans*, or such large emphasis on RGD integrins just because they also evolved early. BTW, the RGD-bearing ligands have changed in evolution, while the laminins have always been the ligands for laminin-binding integrins. The authors seem to talk about RGD binding integrins and all of their hard work as the main story lines, but scientists get credit for their discoveries, not for their work.*

Examples include, but are not limited to, lines 115-119: "Importantly, the two resolved $\alpha 6$ domains (β -propeller and thigh) provide the first atomic view of the integrin α subunit of the laminin binding class. As predicted from the sequence conservation pattern, the seven-bladed β -propeller domain of the $\alpha 6$ subunit bound three Ca^{2+} ions at the bottom of the blades V to VII that are conserved in all integrin subunits (Supplementary Fig. 1), while no Ca^{2+} was observed at the blade IV that is specific for RGD-binding integrins (Supplementary Fig. 1)."

Blade IV that lacks Ca was already observed in crystal structures of the β -propeller domains of $\alpha 4$, αL , and αX subunits and is nothing new.

Lines 120-122: The observations on what is conserved in beta-propeller domains has already been described many times.

We understand that many important structural works have been conducted for non-RGD and non-laminin integrins like $\alpha 4$ integrins and $\beta 2$ integrins, contributing greatly to our current understanding of how integrins generally work. We just tried to emphasize the fundamental importance of laminin-binding integrins in light of their involvement in basic cellular processes since the early metazoans, and to highlight the lack of their structural information in sharp contrast to a wealth of knowledge obtained for other integrin classes. We now have revised the introduction section to incorporate this suggestion. Most of the sentences in the section pointed out by the reviewer regarding the beta-propeller structure (and the Supplemental figures 1a&b associated with it) have been removed.

Point 6(1). Instead of using the hard work and the stepwise progression from crystal to cryoEM as the story line, the paper would be much better written by focusing on structural features, and describing the closed, unliganded and open, liganded structures in parallel rather than in sequence. Thus, how TS2/16 binds would be much better addressed by comparing the two structures. Similarly, the unusual alternatively spliced regions, the subject of the next three points, would be much better described in parallel.

Thank you very much for the suggestion. After testing several different ways of describing each subject in the MS, we decided to describe the crystal structure and the cryo-EM in a successive manner, but tried to streamline the logical flow by focusing on structural features and shortening the crystal structure section so that readers do not have to wait too long until they face relevant issues in the cryoEM structure section.

Point 6(2). The unique features of laminin binding integrins, the X or X1 and X2 regions in their β -propeller domains, which are alternatively spliced in some laminin-binding integrin alpha subunits, get only minor attention but are their most fascinating feature. They are described in lines 123-129, buried at the end of a paragraph with the topic sentence "To gain structural insights into the laminin-integrin interaction, we first aimed at crystallizing human $\alpha 6\beta 1$ integrin ectodomain." They are next described towards the end of Results, lines 254-264. However, their significance is also somewhat unclear, and only with careful reading is it clear that they form part of the ligand binding site! This most interesting feature is also not mentioned in the abstract, where the focus is still on multiple chains in laminin contributing to ligand recognition. One is led not to expect much from this article because the abstract concludes "...laminin recognition by $\alpha 6\beta 1$ integrin follows the "canonical" rule of integrin-ligand interaction, where it involves a direct coordination of a ligand carboxylate to the metal-ion dependent adhesion site of integrin $\beta 1$ subunit, leading to a large-scale shape-shifting of the receptor." (The large shape shifting was already described in an integrin crystal

structure in 2004 for allbb3.) The main point a reader takes away from this MS is how the investigators are real pioneers, and have toiled relentlessly and creatively to complete this work. By contrast, the biological significance of alternative splicing of the ligand binding segment, unique to laminin-binding integrins, is not clearly brought out.

We thank the reviewer again for appreciating the potential importance of the unique feature of the X1/X2 region. We were reluctant to highlight this point too much because it is based on the lack of a (clear) structure rather than a presence of one, and also we did not have much experimental evidence to back the story. Encouraged by this comment, this issue is now mentioned in the abstract and more clearly discussed in the main text with more data obtained by the additional experiments, as explained in the following two answers.

Point 7. Lines 123-125. This reviewer finds the most interesting sentence in the MS here: "We noted that the electron density is particularly weak for the segment spanning K199 to V227 with only the central β -sheet element 211DGPYEV216 showing traceable electron density (Fig. 1c), suggesting a mobile nature of this entire segment including the edge strand of blade III." Whoa, the implications of this are amazing. The Authors are saying that register of the sequence to the structure could shift on the edge strand. But in Fig. 1c it is hard to get the context and realize that this sequence is very close to the laminin binding site. And the text only hints at this possibility and asks the reader to wait until much later in the paper, and the possibility of the register shift is never brought up again. A view of this edge b-strand comes up again in Fig. 4c, but there is no comparison to Fig. 1c, which is distant in the reader's memory here, and the residues in the b-strand in Fig. 1c are shown in stick in Fig. 4c (This reviewer had to read the Figure legend carefully to figure out that this was the edge b-strand, since it was shown all in stick). It is even not made clear if the register is the same in the closed and open structures, and whether there is good experimental evidence for that. Furthermore in that part of the MS, the topic sentence is "Within the integrin-laminin interface, the $\alpha 6$ subunit provides more contacts than the $\beta 1$, contributing toward ~70% of the total interface." BTW, please describe the total solvent-accessible surface buried on each side of this interface.

Much of the importance of the edge beta strand is lost because the Authors discuss the "I-linker loop" in the same paragraph and same figure. BTW, "I-linker loop" is unclear terminology, because its importance in ligand recognition is only in integrins that LACK beta domains. And an I-linker loop is actually one of the two linkers that connect a b-propeller domain to the inserted I domain, not a loop in an integrin that lacks an I domain. It would be much better to use previous terminology such as the blade II-blade III loop, or more concise, the W2-W3 loop or W2 beta4-beta1 loop (W is an abbreviation for blade in the beta-propeller field, as each blade has a W-like topology). Furthermore, the importance of this same loop in ligand recognition has already been described in other integrins, so why change the name in this paper?

We admit that many of our structural drawings lacked careful arrangement to efficiently convey the main message to the viewers. We revised the Fig.1c by changing the orientation of the view and using different color for the X1 region so that readers can understand that the 211-217 segment is in fact the edge strand of blade III sheet. Then in the text we emphasized the fact that the edge strand of blade III is mobile in $\alpha 6$, which is a unique feature not observed in other alpha subunits. Now Fig.1c is viewed from roughly the same orientation with the new Fig.3d, allowing the direct comparison. This presentation also makes it easy to see that the residue register is the same. We also described the buried SASA values on each side in the text (page 8, line 243). As to the terminology of the structural elements, we decided to abandon our former naming (I-linker loop) and simply call it "blade II-III loop" according to the reviewer's suggestion. As each blade of the propeller has four strands, we can call them strands IIIa-III d, which are used to label them in the Fig.1c.

Fig. 4e could be greatly simplified- so many integrins are not needed. The positions of each blade and b-strand should be labeled. Abbreviations such as W3,beta4 could be used. Thus, the important position of W3,beta4, the edge beta-strand where register shifting might occur, could be labeled in the middle of the highlighted X1 region. Even the conventions used and coloring and the description of this figure in the legend need much improvement. The figure legend says the region shown is "near the boundary between blade II and III." This lacks important detail, because the region shown is actually blade III, and includes the boundaries with both blades II and IV. The authors should not use different colors to show residues mutated

in this study or in Ref. 43 and instead should name the residues mutated in the legend. Furthermore, they should highlight residues mutated and found important in ligand binding, not simply “residues mutated.” According to this piece of advice we fully revised the Fig.4e (now it is Fig.3e). The positions for the blade III beta-strands (named IIIa, IIIb, etc as described above) are marked on top of the alignment as arrows, and the actual beta-strand segments in each of the structurally determined integrins (i.e., a6, a5, a4, aV, allb, aL and aX) are underlined. According to the suggestion, we removed a2, a8, and a9, that do not have the experimental structures from the alignment. Legend was also revised, and the residue highlighting scheme was changed as suggested. Additionally, the numbers of the mutated residues are color-coded according to their impact to the binding, thanks to the fact that we now have Kd value for each mutant.

Figure panels 4a-d also need much improvement. There must be a better way to display the laminin-integrin interaction site, including open-book views, possibly two types, one showing interacting residues as sticks with remainder as cartoon, and another as solvent accessible surface, with colors showing regions that are in contact. All easily done with pymol commands such as “select contact, br. A within 4.0 of B” for stick display and omitting “br.” for surface display. Panel 4b is extremely difficult to see, to read labels or see what they label, and needs complete redoing.

According to these suggestions, we fully revised the Fig.4a-d (the new Fig.3a-d). First, the overall laminin-integrin interface (Fig. 3a) is shown as open-book presentation in both surface and cartoon, all using the same coloring scheme as the Fig. 2b. The contacts (within 4Å) are painted in orange in surface presentation and important residues are labeled and shown as stick models in the cartoon presentation. Fig.3b (only this one is made in Chimera) is now drawn as wider shot to make the "column" of $\gamma 1$ tail more easily identifiable, and colors for the residue labels are more carefully chosen.

Point 8. The authors should test the register-shifting hypothesis to bring the biological significance of this article up to Nature Communication standards. These are not difficult experiments. The methodology is straightforward, simple, and already described in Point 4. Highly quantitative measurements can be done of the affinity at equilibrium of laminin fragments to integrins on cells by measuring the concentration dependence of induction by ligand of fluorescent HUTS4 or TS2/16 Fab fragment binding to integrin transfectants using flow cytometry with no washing. As a hint, if the Authors need to increase laminin affinity to save on Fab, they can add an unlabeled extension-stabilizing Fab to the assay. Register shifting is easily achieved by truncating the 194-235 loop stepwise in the W3 beta3-4 loop to shift the W3 beta 4 strand to a more C-terminal sequence register or in the W3 beta4-1 loop to shift the register more N-terminal. The Authors should test the hypothesis that the register can shift to allow laminin-binding integrins to bind to different laminins. The Authors report no register shift when a6b1 binds laminin 511. Therefore, the lowest energy register fits laminin 511, consistent with a6b1 having highest affinity for this laminin. The 194-235 loop contacts the laminin alpha subunit, coded as “5” in the alpha-beta-gamma numbering scheme (abc). The Authors should confirm that other laminins with different alpha subunits bind a6b1 with lower affinity. Then they should test register-shifting deletions. The prediction is that register-shifting deletions would lower affinity for 511, by changing the low energy register position, and by changing the equilibrium toward more N or C-terminal register positions, increase or decrease the affinity for other laminins depending on whether their lowest affinity registers are toward the C or N-terminal positions of the 194-235 loop, respectively. Furthermore, the 194-235 loop should be substituted by the X1, X2, or X1X2 positions of the integrin a3 and a7 subunits (From the text, it was difficult to discern whether a6 has an X2 splice variant, if it does, this should of course also be tested).

The Authors fail to discuss the features of the alpha6 X1 loop that favor a particular register. Y214 and V216 fit into hydrophobic pockets of the b-propeller and are therefore key. The most likely alternative register is achieved when F204 and M206 fit into the same pocket.

If the Authors demonstrate register shifting, it would have enormous importance for the co-evolution of laminins and their cognate integrin receptors from the first metazoans. It would demonstrate a quite different pattern than for RGD integrins, which switched from recognizing one ligand, which disappeared in evolution, to recognizing newly evolving ligands, such as fibronectin. In contrast, laminin and laminin-recognizing

integrins would be shown to have succeeded in co-evolving, owing to alternative splicing (X1 and X2) and register-switching in X1 and X2. Together, these two mechanisms would have provided flexibility for new laminin alpha subunits to evolve, without a loss in recognition by integrins, and with the flexibility to evolve new laminins to fulfill new functions in organisms.

This achievement would not only ensure acceptance by NatCommun, but likely acceptance in a sister journal such as Nature.

As already described in the answer to the Point 4, we have measured laminin-binding affinities for many integrin mutants using fluorescently labeled laminin E8. HUTS-4 (TS2/16 cannot be used because it binds all integrins regardless of the condition as described previously) binding may be used as a surrogate for laminin binding because it only binds to ligand-occupied $\alpha 6 \beta 1$ (Supplementary Fig2d,e), but was not desirable due to the two-step assay format. Direct labeling of E8 fragment with fluorescent dye was considered but was not employed either, due to the potential risk of modifying Lys residues important for the binding. So we turned to GFP fusion with E8 using superfolder GFP that is suitable for extracellular expression, and managed to set up a FACS-based equilibrium binding assay described earlier. This assay was not only more quantitative than the pull-down assay, but also enabled us to evaluate laminin binding to full-length integrin mutants expressed on cells. Results with II-III loop mutants are already mentioned (in the answer to the Point 4). We also measured affinities of X1-region mutants and found several essential (shown in red or orange in the new Fig.3e) and non-essential (green) residues for the interaction. This helped us to give a more accurate picture about the residue-wise contribution to the binding interface, which was not present in the old MS. We next truncated the loop before and after the III_d strand using the loop length of $\alpha 3$ subunit as a guide. The X1- Δ N and X1- Δ C mutants lacked residues 203-209 and 220-227, respectively (boxed magenta in Fig.3e), and X1- Δ NC lacked both. The rationale for making the X1- Δ C mutant was to stabilize alternative residue register (i.e., pulling the F204/M206 to the position of Y214/V216 as suggested by the reviewer). We also hoped that X1- Δ NC mutant may maintain the same strand register as WT, or it may even stabilize this register (and hence increase the affinity?) because the segment has less freedom to shift or detach from the sheet, while long enough to complete the sheet because the length is the same as $\alpha 3$. To our disappointment, these potentially "register-shifted" mutants were all negative in laminin binding, although they were successfully expressed on cell surface judging from the staining with anti- $\alpha 6$ mAb GoH3. As we cannot tell whether these mutants lost the activity because of the altered register or simply because the structural integrity of the entire X1 region was lost, we cannot make strong statements as to the biological meaning of having a mobile III_d strand. We are inclined to think that the mobile nature of this region helps it to explore larger space to transiently capture the LG2 domain using the long-range electrostatic attraction in the initial process of ligand encounter, rather than giving multiple choices of binding platforms. This was our original idea and was already mentioned in the old MS. Regarding the hypothesis that register shift may dictate ligand specificity, we think it is indeed very interesting and thorough investigation is definitely worth pursuing. However, it is out of the scope of this current paper. On top of that, this is a highly original idea of the reviewer him/herself that could not have conceived by us, so we we think we are not entitled to introduce it in our paper.

Point 9. There are multiple criticisms on the structural work for not describing it thoroughly and pushing it further. There is continuous density for the 194-235 loop and so it can be built, for example with PHENIX.ROSETTA which is very helpful at their 3.9 Å resolution. And would also likely greatly improve the fit for the rest of the model, and perhaps for regions of the crystal structure. They should push cryoEM further as described above and describe how they refined. They mention rigid body fitting of both Fv fragments, but obviously refined them as well. They removed HUTS4 Fv for part of the particle registration, but obviously used it in refinement, please describe. Their data cutoffs in Supplementary Table 1 are far too old-fashioned (conservative); they are throwing away much useful data, and need to read and learn modern data cutoff best practices in the paper by Karplus and Diederichs. They fail to show Rwork/Rfree in the highest resolution shell. No results on MolProbity are shown, the use of which in model building (NOT in refinement) could greatly improve their models.

First of all, PHENIX.ROSETTA is designed for the refinement of crystallographic data, particularly for the low resolution data, but cannot be used for EM map data. We did try to build a model for these loop segments

(199-210 and 218-227) as much as we could to the best of our ability, but the resultant models were not convincing, because the model always have one or two residues protruding from the map. So it is clear that, even though the map for this segment is continuous (Supplementary Fig.5c), part of the map is still missing. As for the refinement process, we did perform all-atom refinement using the real-space refinement in Phenix after the rigid-body fitting. This point may have been unclear in the old MS so we revised the Method section. We did not "remove" the HUTS4 densities. Probably the reviewer is referring to the image shown in the middle of Supplementary Fig.3c where the red mask excludes HUTS4 and integrin legs. Masking was intended for the better alignment purpose but they are never subtracted, as the reviewer guessed. We actually tried the density subtraction, hoping that it may improve the quality of the map for the core region containing the laminin-integrin interface. After trying this procedure (called 3D focused alignment and refinement), however, the resolution of the map became worse (4.3Å), forcing us to abandon this result. Although this kind of failed attempts are usually not mentioned in a paper, this comment made us to believe that it would be beneficial to readers to have this information. So we described our attempt and the outcome in the Method section, and revised the Supplementary Fig.3 accordingly.

Based on the suggestion by the reviewer, we re-processed the crystallographic data and in fact succeeded in expanding the resolution to 2.6Å. However, refinement of this new structure resulted in higher R/Rfree value and we could not improve it (see the table below). Thus we had to return to the old 2.89Å data. We did perform MolProbity in the original submission and the results were included in the PDB validation report submitted with the old MS, but we failed to include the results in the table. We revised the Supplementary Table 1 to incorporate the MolProbity result as well as the Rwork/Rfree value for the highest resolution shell.

In summary, this intriguing structural report seems 'under-exploited' with respect to its impact on integrin

X-ray diffraction data collection and refinement statistics 2.89 Å vs 2.60 Å

	2.89 Å	2.60 Å
Data collection		
Resolution (Å)	47.83 – 2.89 (3.07 - 2.89) ^a	47.96 - 2.60 (2.76 - 2.60) ^a
R_{sym}	0.16 (1.23)	0.19 (3.76)
$I / \sigma I$	12.48 (1.63)	9.53 (0.52)
CC1/2	0.997 (0.806)	0.997 (0.349)
Completeness (%)	99.6 (97.7)	99.9 (99.6)
Redundancy	9.8 (9.8)	9.8 (9.9)
Refinement		
Resolution (Å)	44.6 - 2.89 (2.96- 2.89)	44.70 - 2.60 (2.65 - 2.60)
No. reflections	42,218	58,237
$R_{\text{work}} / R_{\text{free}}$ (%)	20.7/24.9 (33.1/39.0)	22.0/25.9 (42.5/45.1)
R.m.s. deviations		
Bond lengths (Å)	0.004	0.008
Bond angles (°)	0.774	0.977
Validation		
MolProbity score	1.74	1.86
Clashscore	8.45	10.97
Poor rotamers (%)	0	0
Ramachandran plot		
Favored (%)	95.90	95.64
Allowed (%)	4.10	4.28
Disallowed (%)	0	0
C β outliers (%)	0	0

recognition by laminins. The authors are urged to perform functional/cellular studies to address key questions that remain unanswered.

Minor issues

-The figures need substantial revisions. Colors are inconsistent - for example, in 1b, the salmon colored domains are 'raspberry' color in 1c. In line 177 of paper, fig 1a is cited, but the text refers to EM images in the literature. The antibody in Fig 1d (cartoon transparency) lies on top of Ca vectors. Also, the colors of

TS2/16 antibody (green/blue) clash with $\beta 1$ subunit (blue). Fig 1e could be greatly simplified - the only difference is the small movement of the α -helix.

We revised these figures accordingly, including simplifying Fig.1d and e (now there is only Fig.1d). We thank this reviewer for these valuable suggestions.

- line 77 - fig 1a citation unusual - refers to EM images

Please refer to our answer to the last comment by the reviewer #1.

- line 99 - replace 'warranted by an aggregation...' by 'mediated by a cluster of small and potentially weak interactions that come from residues distributed across all five subunits'

We changed the text exactly as suggested.

- 113-114 - not clear which parts of $\beta 1$ domain not resolved - is PSI domain at both N/C termini? Figure needs to clarify this

We labeled the positions of the N and C-termini of PSI domain (Q61 and C442) in the Fig. 1c.

- line 124 - difficult to see in Fig1c what they are referring to (residues 211-216)

We revised the Fig.1c inset to simplify the residue labeling (only 211 and 216 is labeled). Also, the main text now refers "Fig.1c, inset" rather than just "Fig.1c".

- line 126 - change 'strange' to 'unusual'. In the current figures, it is difficult to see which parts of β -propeller are unusually mobile, perhaps an additional figure here or in suppl data would help?

Text changed as suggested. The mobile part (the X1 segment) is colored differently in the revised Fig. 1c(as well as in Fig.3d) to make this point clearer.

- 130-132, very long, run-on sentence, please break and simplify.

The sentence was broken into two.

- 144-145, 'and is suggested to be...'

- 146 'an unknown'

The entire paragraph containing these words was deleted, in response to the Points 1 and 2.

- line 327, no contractions (couldn't)

This sentence is no longer there, due to the streamlining of the story lines in response to the Points 3 and 6(1).

Reviewer #3:

Remarks to the Author:

This study provided by Takagi and colleagues reports a cryo-EM and crystallographic structural analyses of $\alpha 6 \beta 1$ integrin alone and in complex with laminin.

Integrins are important cell surface receptors that connect to proteins in the extracellular matrix like laminin and mediate cell adhesion. The presented work revealed how laminin subunits interact with the pocket between the α - and β -integrin heads using a set of conformation-fixing antibodies. Interestingly, using laminin LM511 fragment E8 provided quite a different picture of the interaction compared to published RGD-based ligands or ligand mimetics. It showed a complex binding interface with multiple contact points, and it highlights the uniqueness of the $\alpha 6 \beta 1$ -laminin interaction. Key residues are gamma1E1607 in LM511, which directly coordinates the MIDAS metal in integrin $\beta 1$, and P1609, which contacts the conserved N189 of $\alpha 6$ integrin, in an analogous way as the conventional RGD-based ligand.

This study provides a number of highly relevant residues on laminin and integrin subunits that are key for the fixation of the integrin head opening and therefore, I fully support this study to be published at Nature Communications.

We thank the reviewer very much for the kind and supportive words.

One comment:

In the cryo-EM map, do you actually see the ADMIDAS metal ion? Fig. 4 implies that the metal is there. The authors should clarify this point, as they did not see the ADMIDAS metal ion in the closed integrin without ligand.

Due to the low resolution of the map, the presence of metal ion cannot be definitively confirmed. However, the volume of the map (now shown as Supplementary Fig.4a; it is drawn in Chimera using the contour level of 0.05) suggests that it is not empty. We assume that it is occupied based on this map and the buffer condition (i.e., inclusion of 1mM Mn²⁺ and absence of chelating species). This important point is now described in the revised MS (lines 192-196). Thank you very much.

(End of file)

REVIEWERS' COMMENTS

Reviewer #1 (Remarks to the Author):

None all my concerns were addressed

Reviewer #2 (Remarks to the Author):

This MS is greatly improved. Below this reviewer tries to be as helpful as possible, with a mix of major and minor comments.

Lines 127-137. "Although $\alpha 6$ β -propeller domain assumes canonical 7-bladed propeller fold with the seven four strand anti-parallel β sheets forming the domain core (Supplementary Fig. 1a), we noted that the blade III is somewhat incomplete in $\alpha 6$, with the segment spanning K199 to V227 containing the outermost strand and flanking loops being mostly disordered with only the central β -sheet element 211DGPYEVG217 showing traceable electron density (Fig. 1c, inset). None of the corresponding segment in other published integrin β -propeller domains show this unusual mobility, suggesting the $\alpha 6$ -specific mobile nature of the edge strand of blade III (strand IIIId). Interestingly, this region coincides with the alternatively spliced "X1 region" (residues I193-L235) found only in laminin-binding α subunits 37, suggesting its possible role in ligand capture and/or binding. This point will be discussed later in more detail."

This section is unclear. Central β -sheet elements are central β -strands in β -sheets, not edge β -strands. Also, as the disordered segment does not bind laminin, "specificity" is suggested in place of "binding." Therefore, a possible revision is suggested:

"The $\alpha 6$ β -propeller domain assumes canonical 7-bladed propeller fold with the seven four-strand anti-parallel β sheets forming the domain core (Supplementary Fig.1a). However, the two loops preceding the outermost "d" strand in blade III, i.e. β -strand IIIId, were disordered. In the sequence spanning K199 to V227, only β -strand IIIId, 211DGPYEVG217, showed traceable electron density (Fig. 1c, inset). Density for the large Tyr-214 residue allowed us to confidently assign the sequence to structure register of this segment. No other published integrin β -propeller domains show such long disordered loops. Furthermore, this region coincides with the alternatively spliced "X1 region" (residues I193-L235) found only in laminin-binding α subunits, suggesting its possible role in ligand capture and/or specificity. This point will be discussed later in more detail."

Line 163. Structure determination of the $\alpha 6\beta 1$ headpiece-tLM511E8-TS2/16 Fv-clasp-HUTS-4
Too much jargon for a heading, particularly tLM511E8.

Line 171. Explain meaning of "tLM511E8" and consider changing it for the reader's comprehension. If t is for truncation, is not that redundant with E8? Consider calling it LM511-E8 or simply tLM511.

Line 220 " Consistent with previous epitope mapping result, HUTS-4 recognized species-specific residues E371 and K417 located at the inner face of the lower region of the $\beta 1$ hybrid domain, explaining why it cannot bind to closed $\beta 1$ integrin with tucked hybrid domain (Fig. 2a,b and Supplementary Fig. 5b)".

Please add superposition of closed and open structures in this region. The clash in this region is required to show "why it cannot bind to closed $\beta 1$ integrin with tucked hybrid domain"

Lines 225-226 "by fixing the inter-domain angle." should be "by fixing the PSI-hybrid inter-domain angle."

The figure panels are relatively small in this MS making details hard to see. It is suggested that the layouts be similar to those used in journal pages; in general the aspect ratio should be altered so figures are not tall and narrow, which is true of Figs. 1 and 2. Better to include more figures, Figs. 1 and 2 could be split allowing details to be appreciated. For example, inset Fig. 1c could be larger and

its residues shown as cartoon with only the Y-214 sidechain shown in density to emphasize its role in identifying sequence to structure register. "

Lines 227-239. As TS2/16 is an activating antibody, it must bind with higher affinity to the open than the closed conformation. Although the authors responded to this reviewer's previous suggestion, they did not do the assay that was requested so they will not have pure closed and open populations. However, one expects that in Mg most integrin will be closed and in Mn +RGD most integrin will be open, so an affinity difference would be expected.

Although not noted by the authors, the TS2/16 Fab slides 1 to 2 Å at its interface with the integrin between the closed and open states, which could be sufficient to give a difference in affinity between the two states.

The measurement of affinity for TS2/16 has not been corrected for ligand depletion. The ligand, in this case TS2/16, may be at a concentration near the measured K_d, and a higher affinity for the open conformation could therefore have been missed.

Furthermore, their cryoEM structure is at 3.9 Å and may have biases or insufficient resolution to define the essential differences, which could be small.

Line 241. "Warranted" is the wrong word. Supported or mediated?
Fig. 3a is good but the residues in the contacts should be labeled. This is especially important because it seems the orientation of the cartoon to right is similar but not identical.

Line 255. "we could confirm that γ1E1607 is in fact positioned suitably to form direct coordination to the MIDAS metal (Fig. 3b)" is odd phrasing. It is not positioned in absence of integrin, but is disordered, as laminin-alone structures show. It is better to say. "Our structure shows that integrin binding causes γ1E1607 to become ordered; it is well positioned in the laminin sequence to form a direct coordination to the MIDAS metal (Fig. 3b)"

Lines 258-268 are very well written.

Lines 273-277. Please show an electrostatic surface to confirm this statement - which would require some building, as stated in the response to Lines 456-458.

Lines 362-363. These experiments do not show that the mobility of the X1 region is important, because it was not immobilized. Rather, they show that the mobile X1 region is important.

Line 391-396. There are exceptions to this rule. First, the R is sometimes K, the R and K can be loosely bound, and both Rs and Ks are crystallized in different orientations. Second, structures have shown ligand binding in absence of an R or K. See Lin, F. Y., et al. (2016). "β-subunit binding is sufficient for ligands to open the integrin αIIbβ3 headpiece." J Biol Chem. 291(9): 4537-4546.

Lines 440-448. This is confusing, as X1 is treated as the entire unit that is mobile and its middle portion is structured. Secondly, the loops that are disordered are also disordered when bound to ligand. As the structure could not be built at 3.9 Å in cryoEM, one wonders if it would also appear disordered in the crystal structure. Quite possibly so, as crystal structures at 4 Å or lower resolution are almost never built. On the other hand, it is impossible to conclude that residues 210 and 219 are mobile, as there is density but the loops were not built. Thus, any mobility is highly limited.

Furthermore, truncation of the regions with no essential residues would have changed the overall orientations of the essential charged regions in order to fill the gaps. So this is no evidence that "the long and flexible nature of the entire X1 region may be important" although it does show that its length is important.

lines 456-458. "Based on these considerations, it is tempting to speculate that the long and mobile X1 region is specialized in facilitating the long-range electrostatic capture of ligands, like in a fly fishing."

It is a pity that the authors have not built the disordered regions, which would be required to make an electrostatic surface. They could still build it and not deposit and use it to make an electrostatic surface. One wonders if they deleted end residues before trying to build; Fig. 1c inset and Sup Fig. S4c shows they put D211 sidechain in density where its backbone should have gone and that change alone could have allowed them to build one more residue.

Anyway, it is easy to test if electrostatics are important. All they need to do is a quantitative binding assay at different salt concentrations. Also, electrostatics increases on-rate, so if they also use kinetics, like with SPR or BLI, they can test for the relation between salt concentration and on-rate. There are many examples of papers in which this has been done.

469-487. Concluding paragraph is very nice.

Timothy A. Springer

Reviewer #3 (Remarks to the Author):

The author sufficiently address the questions and comments. I am highly supportive of this impressive study for publication with Nature Communications.

Response to the reviewers' comments

RE: NCOMMS-20-37858-A

Arimori et al.

" Structural mechanism of laminin recognition by integrin"

Followings are our point-by-point responses to the comments and requests provided in the decision letter we received on May 13, 2021. Requests/comments are in *black and italicized* and our responses are in **red**.

Comments by the reviewers

Reviewer #1:

All my concerns were addressed

We thank again the reviewer for taking time to evaluate our MS.

Reviewer #2:

This MS is greatly improved. Below this reviewer tries to be as helpful as possible, with a mix of major and minor comments.

We thank the reviewer very much for appreciating the improvement of the MS, and giving helpful and constructive comments/suggestions again.

Lines 127-137. "Although $\alpha 6$ β -propeller domain assumes canonical 7-bladed propeller fold with the seven four strand anti-parallel β sheets forming the domain core (Supplementary Fig. 1a), we noted that the blade III is somewhat incomplete in $\alpha 6$, with the segment spanning K199 to V227 containing the outermost strand and flanking loops being mostly disordered with only the central β -sheet element 211DGPYEVG217 showing traceable electron density (Fig. 1c, inset). None of the corresponding segment in other published integrin β -propeller domains show this unusual mobility, suggesting the $\alpha 6$ -specific mobile nature of the edge strand of blade III (strand III_d). Interestingly, this region coincides with the alternatively spliced "X1 region" (residues I193-L235) found only in laminin-binding α subunits 37, suggesting its possible role in ligand capture and/or binding. This point will be discussed later in more detail."

This section is unclear. Central β -sheet elements are central β -strands in β -sheets, not edge β -strands. Also, as the disordered segment does not bind laminin, "specificity" is suggested in place of "binding." Therefore, a possible revision is suggested:

"The $\alpha 6$ β -propeller domain assumes canonical 7-bladed propeller fold with the seven four-strand anti-parallel β sheets forming the domain core (Supplementary Fig. 1a). However, the two loops preceding the outermost "d" strand in blade III, i.e. β -strand III_d, were disordered. In the sequence spanning K199 to V227, only β -strand III_d, 211DGPYEVG217, showed traceable electron density (Fig. 1c, inset). Density for the large Tyr-214 residue allowed us to confidently assign the sequence to structure register of this segment. No other published integrin β -propeller domains show such long disordered loops. Furthermore, this region coincides with the alternatively spliced "X1 region" (residues I193-L235) found only in laminin-binding α subunits, suggesting its possible role in ligand capture and/or specificity. This point will be discussed later in more detail."

We fully agree with the criticism and revised the text as suggested (Lines 125-136).

Line 163. Structure determination of the $\alpha 6\beta 1$ headpiece-tLM511E8-TS2/16 Fv-clasp-HUTS-4

Too much jargon for a heading, particularly tLM511E8.

Line 171. Explain meaning of "tLM511E8" and consider changing it for the reader's comprehension. If t is for truncation, is not that redundant with E8? Consider calling it LM511-E8 or simply tLM511.

We apologize for the lack of explanation of the "tLM511E8". It was accidentally removed when we moved large portion of the text to the Supplementary Results. Now the term was simplified as "tLM511" and explained upon the first appearance (line166).

Line 220 " Consistent with previous epitope mapping result, HUTS-4 recognized species-specific residues E371 and K417 located at the inner face of the lower region of the β 1 hybrid domain, explaining why it cannot bind to closed β 1 integrin with tucked hybrid domain (Fig. 2a,b and Supplementary Fig. 5b)".

Please add superposition of closed and open structures in this region. The clash in this region is required to show "why it cannot bind to closed β 1 integrin with tucked hybrid domain"

Thank you for the suggestion. We have made a figure of the HUTS-4-bound hybrid domain superposed onto the tucked hybrid in the closed structure to show the clash (Fig. 4b).

Lines 225-226 "by fixing the inter-domain angle." should be "by fixing the PSI-hybrid inter-domain angle."
Changed as suggested (Line 225).

The figure panels are relatively small in this MS making details hard to see. It is suggested that the layouts be similar to those used in journal pages; in general the aspect ratio should be altered so figures are not tall and narrow, which is true of Figs. 1 and 2. Better to include more figures, Figs. 1 and 2 could be split allowing details to be appreciated. For example, inset Fig. 1c could be larger and its residues shown as cartoon with only the Y-214 sidechain shown in density to emphasize its role in identifying sequence to structure register. "

Thank you for the suggestion. According to this suggestion, we splitted the Figure 1 and 2. Now the Fig.1 contains 4 subpanels, with the former Fig.1c inset being larger and independent subpanel d. We decided to keep all the sidechains for the segment 211-217 to enable comparison with the electron density, but added a label for Y214. The former Fig.1d and e now constitute new Fig.2 a and b, together with the relevant ED map figure for the ADMIDAS region (former Supple Fig.1b). For the former Fig. 2, subpanels a and b were kept to make roughly square-shaped new Fig. 3, and subpanels c and d were moved to new Fig. 4 along with the newly made HUTS-4 superposition suggested by the reviewer as above. As a result, we now have 7 figures instead of the original 5, but the details are much clearer this way and also does not waste too much white spaces when formatted into the Nature Commun style. Thank you very much.

Lines 227-239. As TS2/16 is an activating antibody, it must bind with higher affinity to the open than the closed conformation. Although the authors responded to this reviewers previous suggestion, they did not do the assay that was requested so they will not have pure closed and open populations. However, one expects that in Mg most integrin will be closed and in Mn +RGD most integrin will be open, so an affinity difference would be expected.

Although not noted by the authors, the TS2/16 Fab slides 1 to 2 Å at its interface with the integrin between the closed and open states, which could be sufficient to give a difference in affinity between the two states.

The measurement of affinity for TS2/16 has not been corrected for ligand depletion. The ligand, in this case TS2/16, may be at a concentration near the measured Kd, and a higher affinity for the open conformation could therefore have been missed.

Furthermore, their cryoEM structure is at 3.9Å and may have biases or insufficient resolution to define the essential differences, which could be small.

The reviewer is absolutely right. We looked back our experimental condition and realized that ligand depletion would have indeed occurred! Although the ligand (TS2/16) concentration range was OK, the cell density we used was too high (6×10^6 cells/ml), which calculates into 2 nM $\alpha 5\beta 1$ integrin when assuming presence of 200,000 $\alpha 5\beta 1$ molecules/cell. So we repeated the same experiment with the 30-fold reduced cell density. Sure enough, the binding isotherms became different in the lower ligand concentration range, resulting in the 5~10-fold reduced Kd values from the previous data. More importantly, we did see small but reproducible difference in the affinity toward open and closed integrin this time. So we changed the Supple Figure 5d and its description to incorporate this result. Although this is nice, it does not change the fact that we still do not

know how TS2/16 structurally induce high affinity state of integrin, because the resolution of the cryo-EM is only 3.9 Å, as pointed out by the reviewer. So we rewrote this section as follows (Lines 228-242);

"To our surprise, superposition of ligand unbound structure (X-ray) with the ligand bound form (EM) at the Fv portion of TS2/16 revealed that there was virtually no difference between them on either antibody or integrin side (Supplementary Fig. 5c), indicating that the TS2/16 binding does not cause conformational changes in the β I domain, at least with an extent noticeable at the resolution of the current cryo-EM structure (3.9 Å). As TS2/16 is an activating antibody, theoretical consideration of binding energetics tells us that it must bind with higher affinity to the ligand-bound (open) integrin than the unbound (closed) integrin. Therefore, we compared the binding affinity of TS2/16 toward ligand unbound and ligand-bound α 5 β 1 integrins on K562 cell surface, in a FACS-based binding assay using fluorescent TS2/16 monomer. In fact, the affinity of TS2/16 in the presence of Mn²⁺ and RGD was ~70% higher than that in the resting condition (Supplementary Fig.5d), confirming the theoretical prediction. More precise understanding of the mechanism of the allosteric integrin activation by TS2/16 should await further structural and functional analyses."

Line 241. "Warranted" is the wrong word. Supported or mediated?
Changed as suggested (Line 249).

Fig. 3a is good but the residues in the contacts should be labeled. This is especially important because it seems the orientation of the cartoon to right is similar but not identical.
We considered following this suggestion but decided to keep the current labeling because it will be extremely busy if we label all the contact residues (including ones that merely touches with their mainchain atoms). Actually, the surface (left) and the cartoon (right) are oriented identically. We stated this in the legend (Lines 976-977).

Line 255. " we could confirm that γ 1E1607 is in fact positioned suitably to form direct coordination to the MIDAS metal (Fig. 3b)" is odd phrasing. It is not positioned in absence of integrin, but is disordered, as laminin-alone structures show. It is better to say. "Our structure shows that integrin binding causes γ 1E1607 to become ordered; it is well positioned in the laminin sequence to form a direct coordination to the MIDAS metal (Fig. 3b)"
Changed as suggested (Line 258-260).

Lines 258-268 are very well written.

Lines 273-277. Please show an electrostatic surface to confirm this statement - which would require some building, as stated in the response to Lines 456-458.

We modeled the missing segments and made surface potential figure (Supple Fig. 6b). Please see answer to the "comment to line 456-458" in the next page.

Lines 362-363. These experiments do not show that the mobility of the X1 region is important, because it was not immobilized. Rather, they show that the mobile X1 region is important.
Agreed. We rephrased the sentence as follows. " the mobile X1 region in its entirety is important for the overall binding activity" (Lines 367-368)

Line 391-396. There are exceptions to this rule. First, the R is sometimes K, the R and K can be loosely bound, and both Rs and Ks are crystallized in different orientations. Second, structures have shown ligand binding in absence of an R or K. See Lin, F. Y., et al. (2016). " β -subunit binding is sufficient for ligands to open the integrin α IIb β 3 headpiece." J Biol Chem. 291(9): 4537-4546.
We revised the sentence to incorporate the exceptions and cited the Lin et al. paper. (Lines 396-401)

Lines 440-448. This is confusing, as X1 is treated as the entire unit that is mobile and its middle portion is structured. Secondly, the loops that are disordered are also disordered when bound to ligand. As the structure

could not be built at 3.9 Å in cryoEM, one wonders if it would also appear disordered in the crystal structure. Quite possibly so, as crystal structures at 4 Å or lower resolution are almost never built. On the other hand, it is impossible to conclude that residues 210 and 219 are mobile, as there is density but the loops were not built. Thus, any mobility is highly limited. Furthermore, truncation of the regions with no essential residues would have changed the overall orientations of the essential charged regions in order to fill the gaps. So this is no evidence that "the long and flexible nature of the entire X1 region may be important" although it does show that its length is important.

Based on this criticism, we toned down our speculation on the role of flexibility on the ligand capture by removing most of the lines previous 440-451 segment. (Lines 450-454)

lines 456-458. "Based on these considerations, it is tempting to speculate that the long and mobile X1 region is specialized in facilitating the long-range electrostatic capture of ligands, like in a fly fishing." It is a pity that the authors have not built the disordered regions, which would be required to make an electrostatic surface. They could still build it and not deposit and use it to make an electrostatic surface. One wonders if they deleted end residues before trying to build; Fig. 1c inset and Sup Fig. S4c shows they put D211 sidechain in density where its backbone should have gone and that change alone could have allowed them to build one more residue.

We were reluctant to build models for the loop due to the lack of sufficient information to guide the modeling, but reconsidered it based on this encouragement. We now show putative Calpha models (in different color from the remaining part) for the segments flanking the III_d strand in the new Supple Fig. 4c, and used it to draw surface potential in the new Supple Fig.6b. As the models were built for the presentation purpose only, they were not included in the deposited PDB file and we paid special caution to avoid confusion, by using different color (Supple Fig.4c) or indicating the region with dotted lines (Supple Fig. 6b).

Anyway, the surface electrostatic potential figure (Supple Fig.6b) indeed show highly complementary nature of the surface charge of the interface made by the integrin X1 region (electronegative) and the bottom of the laminin LG2 (electropositive). This figure is used to explain the rationale for the long-range electrostatic capture we claim (Lines 339-349). We are glad that we did this and deeply thank the reviewer for this important suggestion.

Anyway, it is easy to test if electrostatics are important. All they need to do is a quantitative binding assay at different salt concentrations. Also, electrostatics increases on-rate, so if they also use kinetics, like with SPR or BLI, they can test for the relation between salt concentration and on-rate. There are many examples of papers in which this has been done.

Although this is an interesting and important question, it requires setting up another binding experiments using purified integrin and laminin, which could take several months. So we decided not to perform these experiments for the sake of time limitation specified by the editor (2 weeks).

469-487. Concluding paragraph is very nice.

Thank you very much.

Reviewer #3:

Remarks to the Author:

The author sufficiently address the questions and comments. I am highly supportive of this impressive study for publication with Nature Communications.

We thank the reviewer very much for the kind and supportive words.

(End of file)